

# Wind comparisons between meteor radar and Doppler shifts in airglow emissions using field widened Michelson interferometers

Samuel K. Kristoffersen[1], William E. Ward[1], and Chris E. Meek[2]

[1]Department of Physics, University of New Brunswick, Fredericton, New Brunswick, Canada
[2]Institute of Space and Atmospheric Studies, University of Saskatchewan, Saskatoon, Saskatchewan, Canada
**Correspondence:** W.E. Ward (wward@unb.ca)

**Abstract.** Winds from two co-located two wind measuring instruments, a meteor radar and field widened Michelson interferometer at the Polar Environment Atmospheric Research Laboratory in Eureka, Nu, Canada (80° N, 86° W) are compared. The two instruments have very different temporal and spatial observational footprints. ERWIN provides airglow weighted winds from three nightglow emissions (O($^1$S) (oxygen green line, 557.7 nm), an $O_2$ line (866 nm), and an OH line (843 nm)) on a ∼5 minute cadence for measurements at all three heights. As with Fabry-Perot airglow wind observations, these winds are airglow weighted winds from volumes of ∼8 km in height by ∼5 km radius. ERWIN's higher accuracy (1-2 m/s for the O($^1$S) and OH emissions and ∼4 m/s for the $O_2$ emissions) and higher cadence allows more detailed wind comparisons of airglow and radar winds than previously possible. The best correlation is achieved using Gaussian weighting of meteor radar winds with peak height and vertical width being optimally determined. Peak heights agree well with co-located SABER airglow observations. Offsets between the two instruments are ∼ 1 - 2 m/s for the $O_2$ and O($^1$S) emissions and less than 0.3/s for the OH emission. Wind direction are highly correlated with a ∼ 1:1 correspondence. On average meteor radar wind magnitudes are ∼ 40% larger than those from ERWIN. Gravity wave airglow brightness weighting of observations is discussed. Non-quadrature phase offsets between the airglow weighting and gravity wave associated wind and temperature perturbations will result in enhanced or reduced layer weighted wind amplitudes.

## 1 Introduction

Observations of the dynamics and constituent distributions in the mesopause region have increased in importance with recent interest in understanding coupling between the upper and lower atmosphere [Ward et al. (2021); Daglis et al. (2021)]. These observations are relevant to three areas of investigation. The downward transport of $NO_x$ and its subsequent depletion of stratospheric ozone in the polar regions is now known to be dependent on mesopause dynamics [Harvey et al. (2021); Siskind et al. (2021); Sinnhuber et al. (2022)]. A significant amount of the variability in the ionosphere/thermosphere is due to waves passing upward through the mesopause region [Liu (2016); Fang et al. (2018)]. Finally, in itself, the constituent, wave signatures, and





energy and momentum budgets of the mesopause region remain topics of interest [Fritts and Alexander (2003); Eckermann et al. (2018); Swenson et al. (2019); Fritts et al. (2019); Rapp et al. (2021)]. In all three areas, characterization of the dynamics and transport in the mesospause region is essential.

Space based observations of wind in this region have been through interferometric methods (Fabry-Perot, field-widened Michelson and spatial heterodyne spectroscopy) using Doppler shifts in isolated spectral lines in airglow [Shepherd et al. (1993); Hays et al. (1993); Killeen et al. (2006); Englert et al. (2017)]. From the ground, wind observations have been made using lidar [She (2004); Franke et al. (2005); Hildebrand et al. (2017)], radar [Hocking (2011); Stober et al. (2021)], and interferometers [Wu et al. (2004); Shiokawa et al. (2012); Kristoffersen et al. (2013); Langille et al. (2016)]. Radar and lidar are active techniques with information on the atmosphere being determined through the character of radiation reflected from upward-directed generated beams (radio wave or laser). Interferometric techniques are passive with the character of radiation from mesopause airglow emissions being used to determine the atmospheric state. All wind observations are based on Doppler shifts in the observed signal. While the satellite measurements provide global coverage, the details of atmospheric processes on time scales of the order of a day or less can only be obtained through ground based observations [Azeem et al. (2000); Ward et al. (2010)].

Often ground based inter-instrument comparisons are used to validate new observing techniques. Since the manner in which each instrument technique provides information on the atmosphere (termed the instrument filter) is different, multi-technique observations provide richer data sets than single instrument observations. Processes linking variations in wind, temperature, density and constituents over a range of spatial and temporal scales can be investigated. Data from an early multiple technique campaign [Swenson et al. (1995)] used lidar, airglow imagers and a Michelson interferometer to investigate the character of waves in the mesopause region. Recognition that various instrument techniques sample different parts of the gravity wave spectrum followed [Gardner and Taylor (1998); Franke et al. (2005)]. Further multi-instrument studies (see for example Tang et al. (2002); Fujii et al. (2004); Ejiri et al. (2009); Suzuki et al. (2010)) occurred in the next decade and some of the practical details involved in such studies were addressed. Recently more ambitious observation campaigns have taken place involving multiple-instrument, multiple site and satellite observations, as well as model validation and data assimilation [McCormack et al. (2017); Eckermann et al. (2018); Reichert et al. (2019); Vargas et al. (2021); Stober et al. (2021); Rapp et al. (2021)].

One multi-technique combination whose potential is being developed is airglow interferometry and meteor radar. Comparisons between the techniques have a long history, with earlier work being primarily concerned with the validation of the two techniques [Burrage et al. (1996); Gault et al. (1996); Meek et al. (1997); Plagmann et al. (1998)]. An issue for these comparisons was that the height of the airglow layer varies significantly [Swenson et al. (1989); Ward et al. (1994); Plagmann et al. (1998); Zhao et al. (2005); Marsh et al. (2006); Liu and Shepherd (2006)] so the airglow winds could not be associated with a fixed height. Ward (1999) clearly laid out the physical mechanism behind this variability and Vargas (2019) has summarized





confidence intervals for various derived quantities associated with height uncertainties.

Recent papers have compared variations in Doppler airglow winds with meteor radar winds, and estimated heights of the layer [Yu et al. (2017) and Lee et al. (2021)]. In this paper, we continue to investigate the character of these correlations, albeit with a field-widened Michelson interferometer, the E-Region Wind Interferometer II (ERWIN) and a co-located meteor radar (SKiYMET) [Hocking et al. (2001)]. The ERWIN and MWR winds used in this study are from the December 2017, and January 2018 period.

The basic measurement process for the field-widened Michelson is the same as for the Fabry-Perot (both use Doppler shifts from isolated spectral lines in airglow [Burrage et al. (1996); Fisher et al. (2000)]). However, ERWIN has a significantly faster temporal cadence and wind accuracy than the Fabry-Perot [Kristoffersen et al. (2013)]). These enhanced observation capabilities allow the relationship between wind measurements with the two techniques to be explored in more detail than previous analyses. Establishing the complementary relationship between these two techniques provides a foundation for future enhance-
ments in ground based observations of wind and constituent transport.

This paper is organized as follows. Following this introduction, the instrument capabilities, the observation site and the instrument filter associated with each instrument are described. The character of the wind observations from the two instruments and the analysis of various correlation strategies comprise the content of the next section. The interpretation and implications
of these results are then discussed. The final section summarizes the main results of the paper and future research directions.

## 2    Instruments

The instruments used in this study, the E-Region Wind Interferometer II (ERWIN ) [Kristoffersen et al. (2013)] and an All-Sky Interferometric Meteor Radar (SKiYMET) [Hocking et al. (2001)], are co-located at the Polar Environment Atmospheric
Research Laboratory (PEARL; Eureka, Nu, Canada: 80° N, 86° W). These are two of a suite of instruments at PEARL measuring constituents, aerosols, temperature and wind from the ground to the thermosphere [Drummond and Team (2017)]. They implement very different methods of wind measurement. ERWIN uses Doppler shifts in airglow emissions and the SKiYMET radar reflects radio waves off meteor trails.

Wind measurements with ERWIN are through optical interferometric means. They consist of airglow radiance weighted integrated line-of-sight winds with a small spatial cross section. Airglow consists of naturally occurring molecular and atomic emissions, which, in the MLT region at night, is associated with three body reactions involving atomic oxygen [Slanger and Copeland (2003)]. Because of the height dependence of three-body reactions and the increase in quenching rates at lower altitudes, the height profile of airglow emissions is quasi-Gaussian shaped with half-widths of ∼8 km. In contrast, the meteor





radar measures winds using a temporal average of Doppler shifted echoes of radio waves from spatial regions, which typically have vertical thicknesses of a few kilometers and a horizontal scale of a few hundred kilometers.

As the two techniques provide winds in the same altitude range through two independent and different approaches, consideration of the instrument filter of each is important in determining how to combine them [see Gardner and Taylor (1998); Franke

et al. (2005)]. The spatial characteristics of the wind observations and the temporal and spatial details of the two techniques is summarized in Figure 1. Of particular note is that ERWIN provides airglow weighted zonal (meridional) wind measurements from line of sight observations in the east and west (north and south) from cylindrical volumes with radii of ∼5 km and lengths of ∼8 km at a temporal cadence of ∼5 minutes. The meteor radar typically provides an hourly vertical profile of horizontal wind measurements with height bins of 3 km averaged over a circular disk of ∼300 km. Details of the two techniques is pro-

vided in the following two subsections.

## 2.1  Michelson Interferometer

ERWIN was installed at PEARL in 2008, and has been operational since that time. It is a Doppler Michelson interferometer, which measures airglow irradiance and wind using Doppler shifts in three airglow emissions: the oxygen green line (O($^1$S)

henceforth) at 557.7 nm, the $^P$P(7) and $^P$Q(7) lines in the $O_2$(0–1) atmospheric band at 859.9 and 860.0 nm ($O_2$ henceforth) and the P$_1$(3) emission line in the OH(6,2) Meinel band at 843.5 nm (OH henceforth). The nominal heights of the emission peaks are ∼97, 94 and 87 km and the wind accuracies are ∼1.5 m/s, 4 m/s and 1.5 m/s respectively. The measurement cadence is  5 minutes for line-of sight wind measurements in the four cardinal directions and the vertical for all three emissions. The five measurement locations are zenith, and each of the four cardinal look directions at 51.3 degrees from zenith. Given the

nominal heights of the airglow layers, the wind observations in the cardinal directions are located horizontally, approximately 125 km from the zenith measurement.

The details of ERWIN including its operations are covered in Kristoffersen et al. (2013). Here we provide a short summary of the essential points. Wind measurements for each emission, are made by simultaneously imaging light through the inter-

ferometer from all four cardinal directions and the vertical onto different sectors of the charge coupled device (CCD). This avoids the need for a turret to sequentially view each direction as is often implemented in other airglow wind instruments. The interferometer is then stepped through 8 steps to sample the resulting fringe and determine phase changes relative to a zero wind background. Doppler shifts from each illuminated bin on the CCD are then calculated from these phase shifts and averaged over each segment to determine line-of-sight winds in each direction. Zonal and meridional winds are then determined

by subtracting the west from the east and south from the north respectively. This procedure eliminates the contribution of the vertical wind to the line-of-sight winds assuming it is homogeneous across the field.





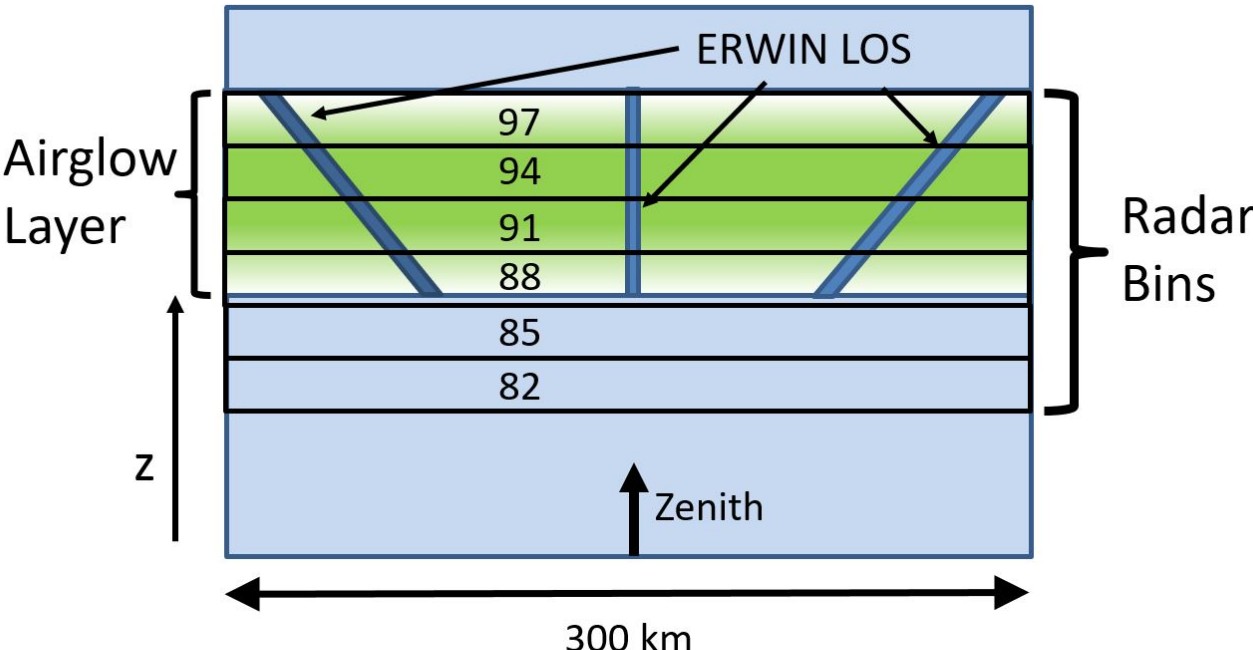

**Figure 1.** A schematic showing the spatial characteristics of ERWIN and the meteor radar observations followed by a table summarizing the temporal and spatial parameters of the two techniques



Typically the order of operation is to sequence through the emissions, making wind determinations followed by the corresponding calibration, for each airglow emission. The calibration consist of a measurement of fringe phases of isolated emission lines from noble gas microwave discharge lamps close in wavelength to the airglow emissions. These allow any slow temporal variations in the the observed Michelson fringes caused by changes in ambient environmental conditions to be monitored and accounted for. They also allow absolute winds in each direction to be determined with the assumption that the vertical wind averages to zero over a day.

Given the simultaneous observations of the four cardinal line-of-sight wind directions, it is possible for the winds to be self-validated. By considering winds averaged over a period of a day or longer, the opposing directions (e.g. north and south) would be expected to have the same magnitude wind values, but with the opposite sign. This was demonstrated in Kristoffersen et al. (2013) for several different days. These comparisons of the opposing winds demonstrate that there is no systematic offset in the meridional or zonal winds.

The uncertainties in these wind measurements were determined by assuming that each line-of-sight sector should provide a single value for the wind. Given this assumption, the line-of-sight wind uncertainty is simply the standard error of the roughly 140 bin quadrants, with the line-of-sight wind value being the mean of each quadrant Kristoffersen et al. (2021).

## 2.2 Meteor Wind Radar

The SKiYMET meteor radar (hereinafter referred to as the meteor wind radar or MWR) is co-located with ERWIN at the PEARL observatory and was installed there in 2006. It is an implementation of the well known meteor radar wind measuring technique [Hocking et al. (2001); Holdsworth et al. (2004) which uses Doppler shifts in radio waves reflected off meteor trails to determine winds in the mesopause region. The measured range, consisting of 30 2 km gates, is over sampled; the correct range alias being selected according to whether the resulting calculated height (from range and angle of arrival, AOA) is in the expected meteor region. Meteors with ambiguous range or AOA, (i.e. those not within 10-70 degrees zenith angle, and those with estimated relative errors in radial velocity greater than 25% are rejected. Standard processing is done in in 90 minute intervals, shifted by 1 hour, at six ~3 km layers centred on 82, 85, 88, 91, 94, 97 km.

The velocity fit is a two-pass process. The first deletes outlier meteors. The refit requires at least 7 remaining meteors. Since the measurements are radial, meridional/zonal component uncertainties are generally not available because a perturbation in radial velocity due to one wind component affects both components in the analysis. In addition, meteor distributions are not uniform in azimuth (or zenith), so one component may be better determined than the other. With the standard processing parameters and by differencing adjacent wind values, an estimated uncertainty of about 5 m/s in each of North and East components was found for the middle four heights. The uncertainty is greater at 82 and 97 km, where fewer meteors are detected.





For comparisons where the vertical resolution was of particular importance, binning at 1 km height bins was also calculated. The uncertainties for this binning were greater so these were not generally used since the statistics of the correlations were of more importance. Where the higher vertical resolution observations were used is clearly indicated.


## 3  Preparation of Suitable Data Products

To properly compare the winds measured by these two instruments, data products of a comparable nature must be derived from each instrument. To account for the differing instrument filters involved, the ERWIN observations were averaged using a similar averaging algorithm as the MWR (i.e. using a 90-minute running mean, centred on the hour, stepped by an hour
to provide hourly wind measurements (zonal and meridional) on the same temporal cadence as the MWR). As ERWIN is an optical instrument, the presence of clouds obstructs the airglow emissions, and effectively acts as a diffuser Langille et al. (2016); Kristoffersen et al. (2021). The resulting diffuse light, no longer has accurate wind information so cloudy days were removed from the data set, so that it only includes days when the observed airglow winds accurately represent the neutral winds in the mesosphere. Cloudy periods were determined using information from a co-located Millimetre Cloud Radar operated by
Environment and Climate Change Canada.

Careful attention must be paid to the determination of the uncertainties in any quantity derived from the basic wind measurements of each instrument. This was undertaken by calculating the wind variances associated with each instrument in a similar manner. As ERWIN is the instrument with a higher temporal cadence, geophysical variability in the winds was defined
as the variance of the ERWIN winds over each 90-minute window divided by the number of measurements in the window (i.e. essentially the standard variance). This variance was then added in quadrature to the variances associated with the instrumental uncertainties of each instrument. The estimated MWR uncertainties noted above were used for the meteor wind instrumental uncertainties. For ERWIN, the mean variance over the interval was determined using the uncertainties associated with each wind observation. It was found that the instrumental variances in both cases were much smaller than the geophysical variances
and as a result, the two instruments had very similar variances.

These derived data products form the foundation for the comparisons and analyses which follow and allow issues such as height variations in the airglow layer to be explored.

## 4  Wind Comparisons

There are several issues to consider when comparing the winds from these two sets of observations. As noted above, the instrument filters differ with ERWIN providing airglow weighted winds over a region with a diameter of ∼5 km and length of ∼8





km (for the horizontal winds) and the meteor radar providing winds averaged over a disk of 3 km height and ∼300 km radius. Furthermore, the height of the airglow layer can vary by several kilometers [Ward (1999)] so the height of best correlation will vary in time. Finally, the airglow layers for the O($^1$S) and $O_2$ emissions extend above the heights sampled by the radar so these comparisons will be less robust than comparisons with OH.

The winds from the two instruments are compared in several different ways. The first is a simple single height comparison using the full data set, which provides a reference for the correlation coefficient and the linear fit parameters. The second is a best fit Gaussian weighted average of the MWR winds over a range of heights, i.e. vertically averaged meteor radar observations, again using the full data set. Height variations in the airglow layer are explored by considering short term variations in the height of best correlation. This is accomplished by finding the time series of Gaussian weighted (i.e. variable height and width) MWR observations which best correlated with the ERWIN observations for each two-day interval and constructing a running mean of these heights for the observation period. Finally, correlations in terms of wind direction and magnitude are considered.

The comparisons are made through consideration of the correlation coefficient and the form of a least-squares fit to a linear model. Ideally, the correlation coefficient should be close to 1, the slope of the linear model ∼ 1 and the intercept ∼ 0. The closeness of these parameters (in particular the correlation coefficient) to these ideals is used to evaluate the quality of the comparison.

Figure 2 shows wind time series of the MWR winds for each measured height, along with the ERWIN winds at the nominal airglow layer heights for the data set used in this paper. Wind vectors from the meteor radar are plotted in black and those from ERWIN in red, with periods where there are no ERWIN winds left blank. The ERWIN winds generally complement the MWR winds well, with some discrepancies due to the differences of the height at which each measurement is made.

To provide a baseline against which various comparison methods can be evaluated, direct correlations between the ERWIN winds and MWR winds at each height for the full time series of the data set were calculated. Examples of the correlation plots of the ERWIN winds vs the MWR are presented in Figures 3 and 4. They provide information about how well these winds compare - not shown for conciseness are the corresponding figures of the zonal wind correlation plots and with lower correlation coefficients (due to the reduced precision of these winds), all those involving the $O_2$ winds. The first figure is a correlation scatter plot of the O($^1$S) meridional winds and the MWR winds at six of the measured heights, and the second a similar figure for the OH meridional winds and the MWR meridional winds.

The value of the correlation coefficient varies with height. In the case of the O($^1$S) winds, the correlation increases up to 94 km, and decreases at 97 km. This does not correspond to the nominal height generally associated with this emission. This is likely because the meteor radar detects few meteors above 94 km so winds at 97 km are less reliable than at lower heights. In



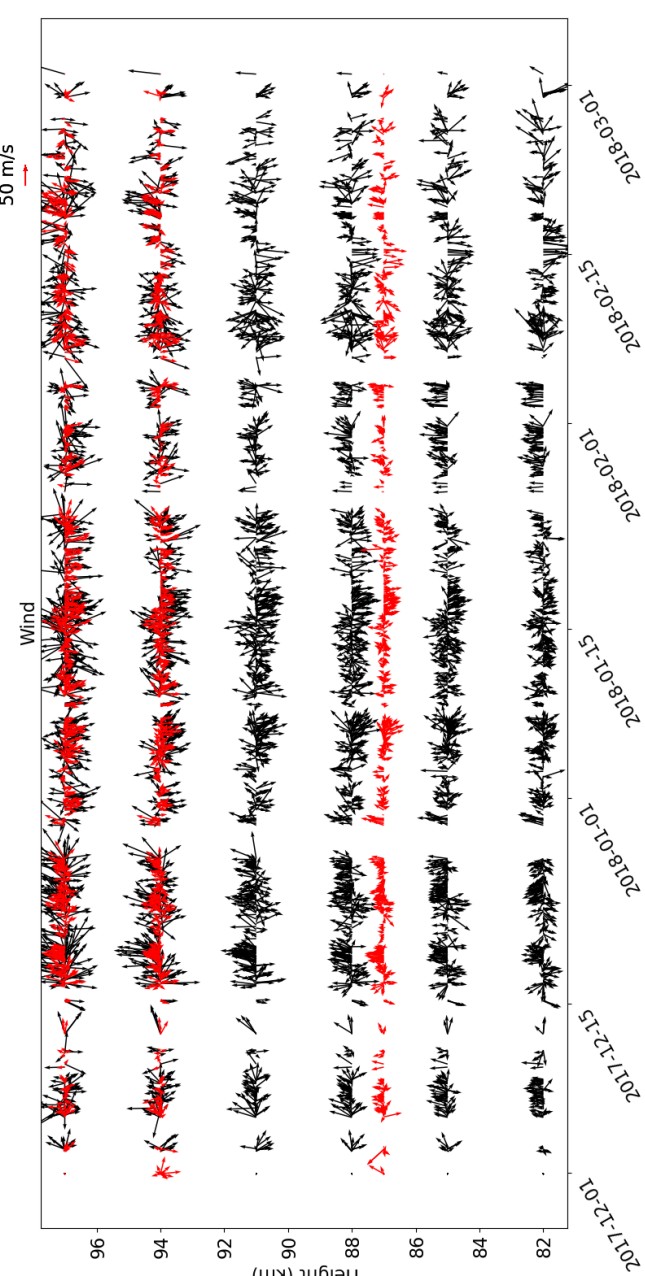

**Figure 2.** Arrow plots of the MWR (black) and ERWIN (red) wind vectors. The ERWIN winds are shown at the nominal emission heights, such that the O($^1$S) is 97 km, O$_2$ is 94 km, and OH is 87 km. Times when there were no ERWIN measurements are left blank.





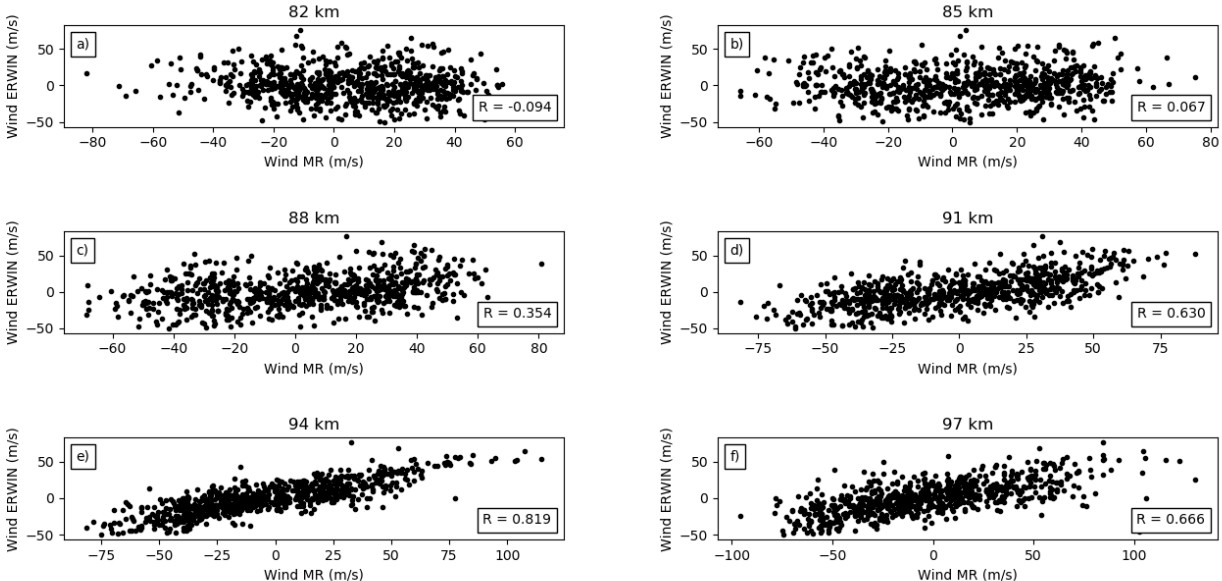

**Figure 3.** Correlation scatter plots of the meridional component of the O($^1$S) ERWIN winds (y-axis) and MWR winds (x-axis) at a) 82 km, b) 85 km, c) 88 km, d) 91 km, e) 94 km, f) 97 km.

addition, there is significant O($^1$S) emission above 97 km and winds from these heights are not observed by the radar. The OH correlations maximize at about 88 km in close agreement with the nominal height of the OH airglow layer of 87 km. In similar

plots (not shown) for the O$_2$ wind and MWR, the correlation coefficient peaks at 92 km, again in reasonable agreement with its nominal height of 94 km.

Figure 5 summarizes these correlations for the three emissions for both meridional and zonal winds. For this figure the higher vertical resolution MWR observations were used. Each correlation curve has a roughly Gaussian shape, with the maximum

correlation occurring at heights ∼95 km, ∼91 km, and ∼87 km, for the O($^1$S), O$_2$ and OH emissions, respectively. The curves are similar for both meridional and zonal winds. These are close to the nominal heights for these emissions (although a little low for the O($^1$S) likely for the reasons discussed above). These results support the use of correlations between the airglow weighted winds and the MWR winds to obtain information on the height and shape of the airglow layer.

Figure 6, contains panels of the correlation plots for the height where the correlations maximize. In each plot, the R-value associated with the correlation and the parameters associated with the best linear fit (assuming errors in both data sets) are listed. The R-values are high, between 0.77 and 0.88 indicating excellent correlation between the MWR winds, and the ERWIN winds. The slopes range between 0.62 and 0.77 indicating that the wind variation is larger in the MWR winds. The intercepts





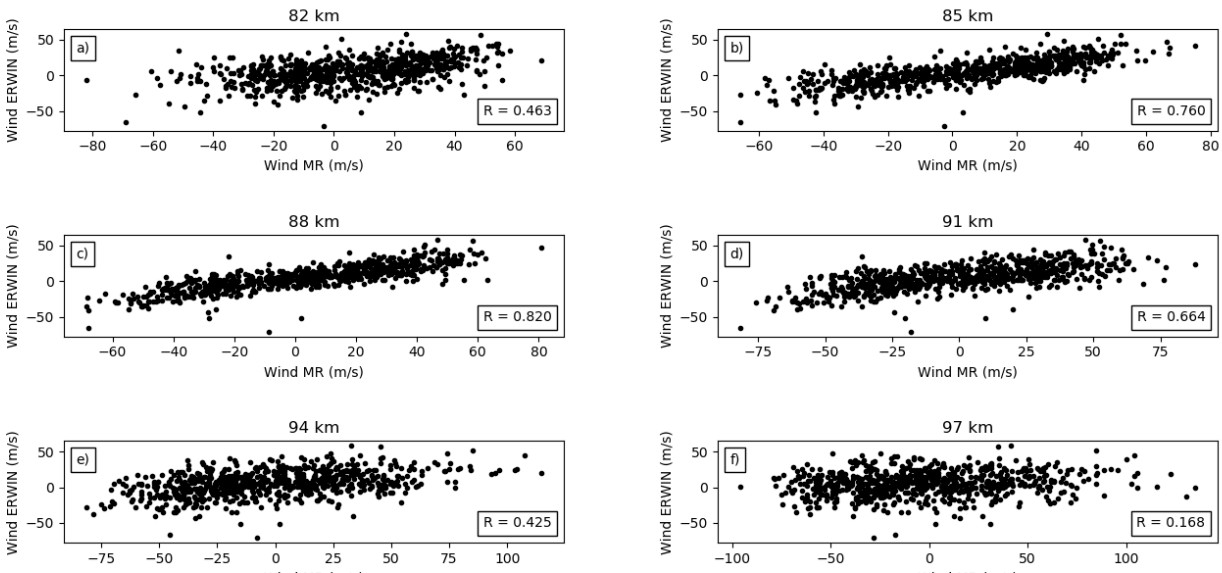

**Figure 4.** Same as Figure 3 but for the OH emission ERWIN winds.

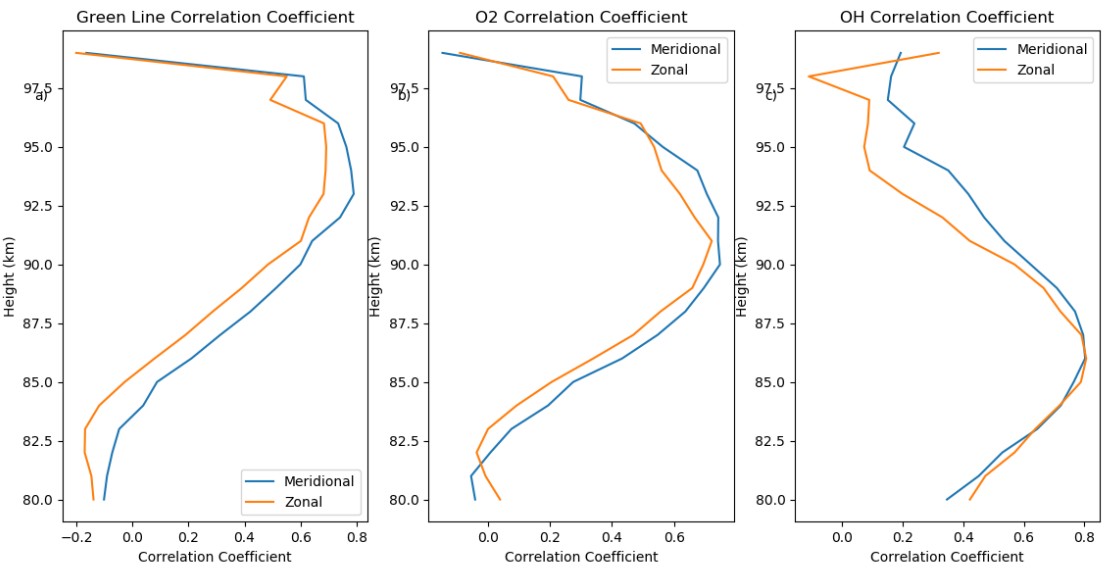

**Figure 5.** Correlation coefficients as a function of height for O($^1$S) (left panel), O$_2$ (centre panel), and OH (right panel) emissions. The meridional and zonal wind correlations are the blue and orange lines, respectively.





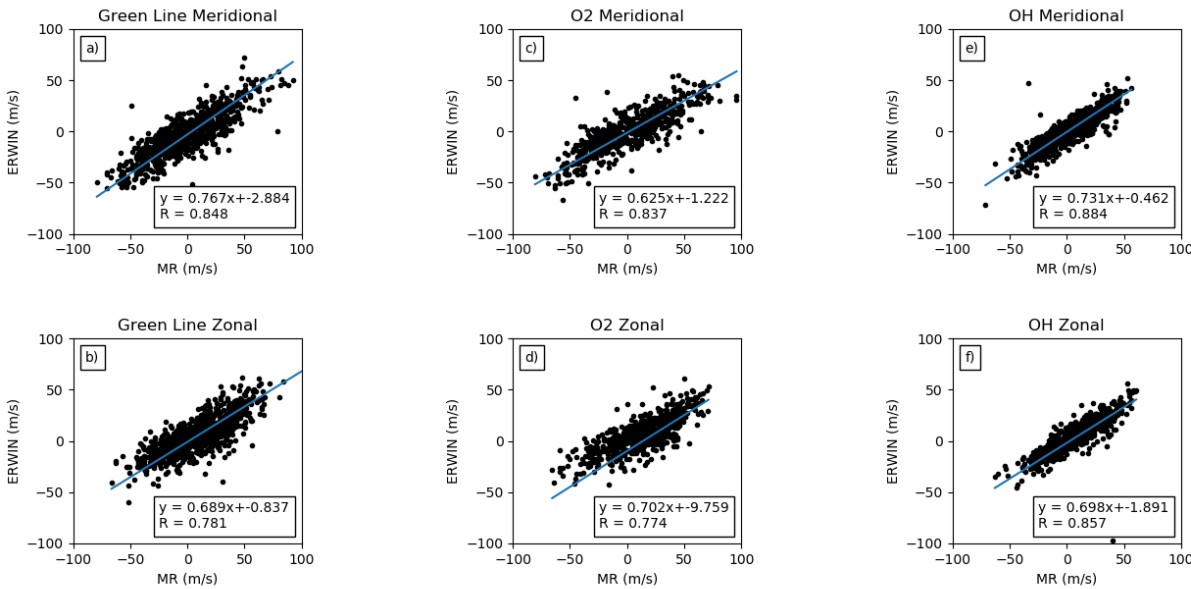

**Figure 6.** Correlation scatter plots for the ERWIN winds and MWR winds for the height of best correlation for each emission. The results for the meridional winds are on the top row and those for the zonal winds on the bottom row. The O($^1$S) emission results are in the left column, the O$_2$ results are in the centre, and the OH results in the right column. The equation for the best linear fit and the r-value of the correlation are in the lower left of each panel.

for the fits are negative and range between -0.462 and -2.884 except for the O$_2$ zonal wind which is -9.76.


The airglow winds are brightness weighted along the line-of-sight. Hence, single height comparisons are an over simplification. A more representative measure of the wind correlations is to use an average over multiple heights of the MWR winds. To approximate the effect of the brightness weighting by the airglow layer the meteor winds are weighted in height with a normalized Gaussian. This approximates the volume emission height variation of an airglow layer. The peak height and full-width at

half-max (FWHM) were varied and the best correlations for each emission, and direction (meridional and zonal) determined. The FWHM was calculated from the standard deviation associated with the best Gaussian fit. The results are provided in Figure 7.

For this set of correlation calculations, the height/width combinations with the highest correlations provide information on

the height, and thickness of the airglow layers although this must be interpreted carefully (see below). For all emissions, the peak height and width of the Gaussian which resulted in the best correlation were the same for the meridional and zonal winds. For the O($^1$S) calculations, the highest average correlation (i.e. an average of the zonal and meridional correlations) $\bar{R} = 0.818$, was achieved for a Gaussian distribution with a peak height of 94 km, and a FWHM of $\sim$ 5 km. For the O$_2$ emission the highest





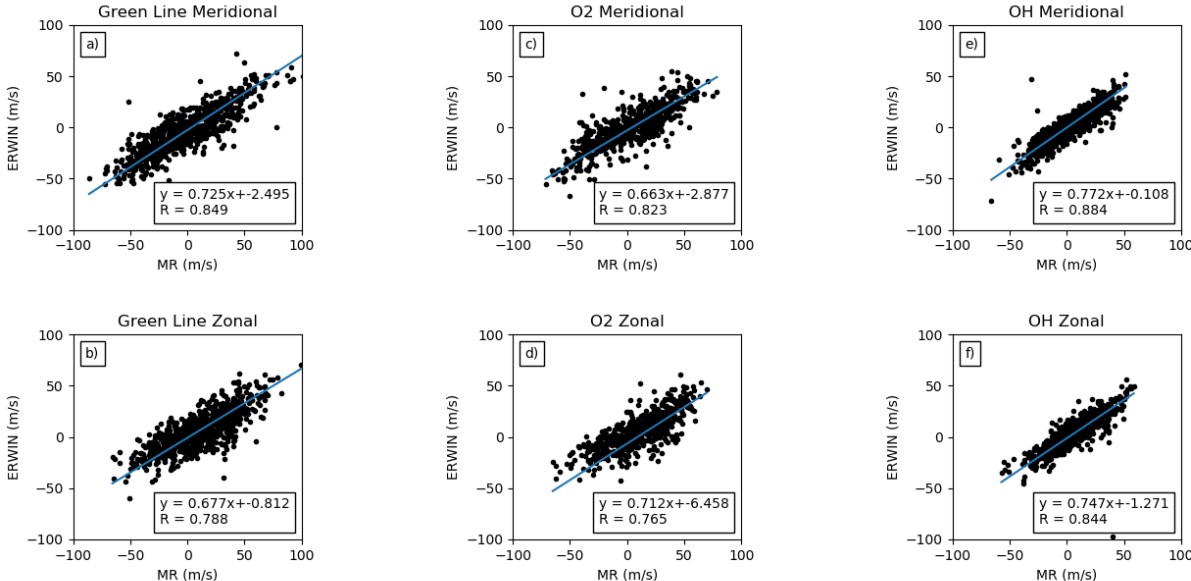

**Figure 7.** Same as Figure 6 but showing the best correlations using the Gaussian weighted height averages of the MWR winds at the height of best overall correlation.

average correlation was $\bar{R} = 0.794$ for a peak height at 91 km and FWHM of $\sim 5$ km. For the OH emission, the highest average

correlation was $\bar{R} = 0.864$ for a peak height of 85 km and a FWHM of $\sim 14$ km.

Relative to the baseline single height correlation calculations (i.e. Figure 6), the correlation coefficients increased slightly for the $O(^1S)$ emission, decreased slightly for the $O_2$ emission and the OH zonal calculation and remained equal for the OH meridional calculation. The slopes of the best linear fit increased for the $O_2$ and OH analyses but decreased for the $O(^1S)$

analysis.The magnitude of the offset increased slightly for the $O(^1S)$ analysis but decreased for the $O_2$ and OH zonal emission analysis and remained the same for the OH meridional analysis.

A limitation of the previous analysis is that the correlation analysis involves the full time series (two months). Variations in the height of the airglow layers are not accounted for and will decrease the correlation at a given height. To improve these

correlations, and to get a sense of how the airglow layer height is varying in time, further analysis was undertaken. This analysis involved correlating the ERWIN and MWR winds over two-day periods, stepped by a day. The height of the strongest correlation at each two day period generates a time series of layer heights and provides an indication of the variability of the peak airglow heights with time. For this analysis the MWR observations on the 1 km height grid were used.




Figure 8 presents the two-day correlations between the ERWIN winds and the MWR winds for the Dec 1, 2017 through Jan 31, 2018 time period. The height of highest correlation for each emission is denoted by an 'x'. Since the Sounding of the Atmosphere using Broadband Radiometry (SABER, Russell III et al. (1999)) was measuring two OH bands and $O_2(^1\Delta)$ airglow emissions near Eureka in January, 2018, the associated peak emission heights are also plotted for comparison (the red and pink coloured '+' for OH and the red coloured '+' for $O_2$). Note that since the $O_2$ emission that the SABER observes is

different from the one observed by ERWIN, $(O_2(^1\Sigma))$, it generally peaks at lower altitudes [Noll et al. (2016)] as seen in this figure . The SABER peak airglow height values were determined from daily averages of observations in the latitude range of 77N to 83N degrees north latitude and 255 to 293 degrees east longitude using Version 2.07 of SABER data.

       In this figure, the peak emission height determined from the the meridional and zonal wind ERWIN/MWR correlations

match well for the three emissions. The basic pattern of emission height variation appears in all three emissions suggesting that they are associated with common large scale dynamical events. In addition, the match to the SABER peak height determinations is good. The SABER match validates the use of this correlation approach to explore variations in the airglow peak height. Variations of close to $\pm$ 5 km are observed in the OH and $O_2$ analyses. As the nominal $O(^1S)$ emission peak is close to the top of the MWR observations (this consideration also applies to a lesser extent to the $O_2$ observations), it is unlikely that the full

extent of the variations with this emission would be observed, but even here the total variation is close to 10 km.

       Comparisons of the ERWIN and MWR meridional and zonal winds determined using this running mean approach appear in Figure 9. The agreement is good with the two time series matching on average. The MWR winds vary over larger ranges than the ERWIN winds do.


       Scatter plots using these two-day running mean optimal heights to define the Gaussian averaged MWR winds were calculated and compared to the ERWIN airglow winds (see Figure 10). Relative to the baseline single height correlations, the correlation coefficient increased for all cases except for the $O_2$ meridional wind comparisons. Similarly, the slopes for all cases decreased except for the $O_2$ meridional wind comparison where it increased. For the OH comparisons, the magnitude of the

offset decreased to less than 0.3 m/s for both components. For $O_2$ it decreased for the zonal component and increased for the meridional component. Finally for the $O(^1S)$ it decreased slightly for the meridional component and increased slightly for the zonal component.

       Another way to examine the relationship between the two data sets is to compare the magnitude and direction of the two

data sets. The time series of ERWIN and the 2-day running mean Gaussian average meridional and zonal winds were converted to magnitude and direction time series and compared. The resulting scatter plots are in Figure 11. In this plot the upper panels correspond to the wind magnitude and the lower panels to the direction. From left to right, the columns correspond to the $O(^1S)$, $O_2$ and OH emission analyses. The red lines correspond to the best linear fit to the observations and in the plots



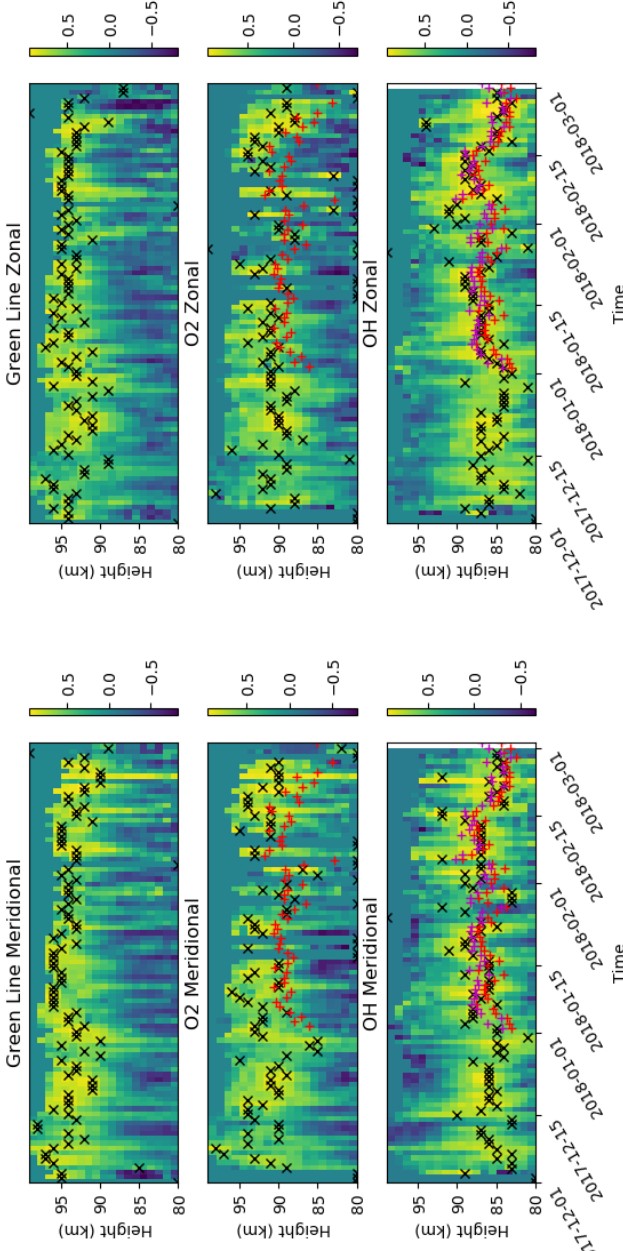

**Figure 8.** 2-day correlation coefficients for comparisons between the MWR and the ERWIN winds are shown as a function of the height (y-axis) and time (x-axis). The meridional and zonal winds are shown in the left and right columns, respectively. The O($^1$S) emission are the top row, the O$_2$ emission is the middle row, and the OH emission is the bottom row. The heights of maximum correlation are denoted by the 'x's, with the O$_2$ and OH emission heights from SABER are denoted by the '+'s. The red '+' on the middle panels are the SABER O$_2$($^1\Delta$) heights. The red and magenta '+' on the bottom panels are the SABER OH 2.0 $\mu$m and 1.6 $\mu$m emissions, respectively.





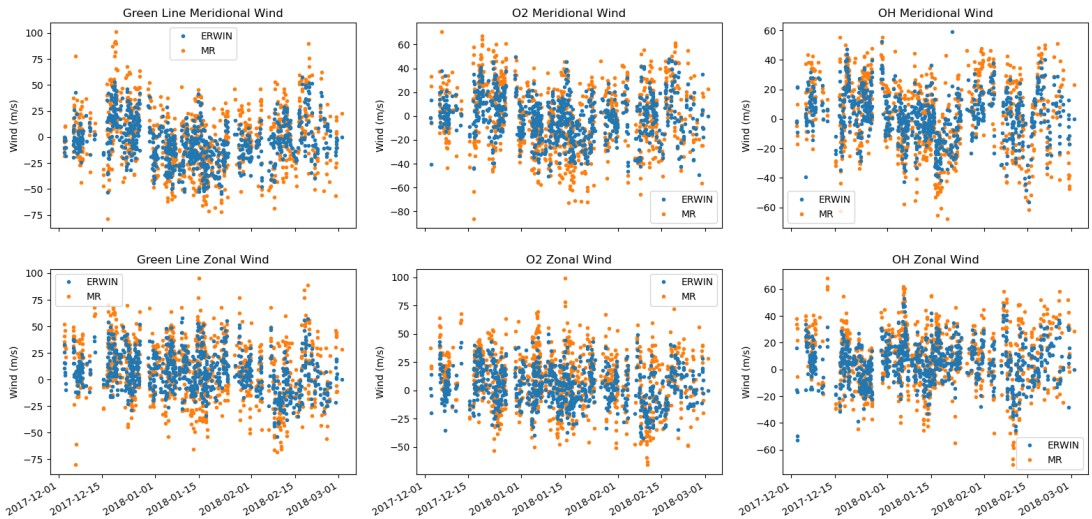

**Figure 9.** Plots of the ERWIN winds (blue) and 2-day best height, Gaussian-weighted MWR winds (orange). The meridional and zonal winds are shown in the top and bottom panels, respectively. The left column is the O($^1$S) wind, middle column is the O$_2$ wind, and the right column is the OH wind.

associated with the directions, the black line corresponds to a line with a slope of 1 passing through the origin.


This figure indicates that the main difference between the two data sets lies in the magnitude of the winds. The respective O($^1$S), O$_2$ and OH correlation coefficients for the wind magnitude analyses are 0.698, 0.632 and 0.72 and significantly higher - 0.860, 0.875 and 0.921 - for the analyses of the directions. The respective slopes of the linear best fits are 0.619, 0.572 and 0.701 for the magnitude fits and 0.932, 0.987 and 1.023 for the fits to the direction. The intercepts for the magnitude fits are

1.923, 3.113 and -1.175 m/s and for the directional fits, 0.007, 0.067 and 0.061 respectively. ERWIN on average measures smaller wind magnitudes than the MWR.The wind directions are highly correlated with a 1:1 relationship and close to zero intercept.

## 5   Analysis

Several approaches to comparing the ERWIN and MWR were presented in the previous section. Table 1 summarizes the resulting parameters for the three correlation types. For each emission and wind direction, the "optimal" parameters for r-value, slope and intercept are underlined and bold faced (i.e. highest correlation parameter and linear fits closest to a slope of 1 and a




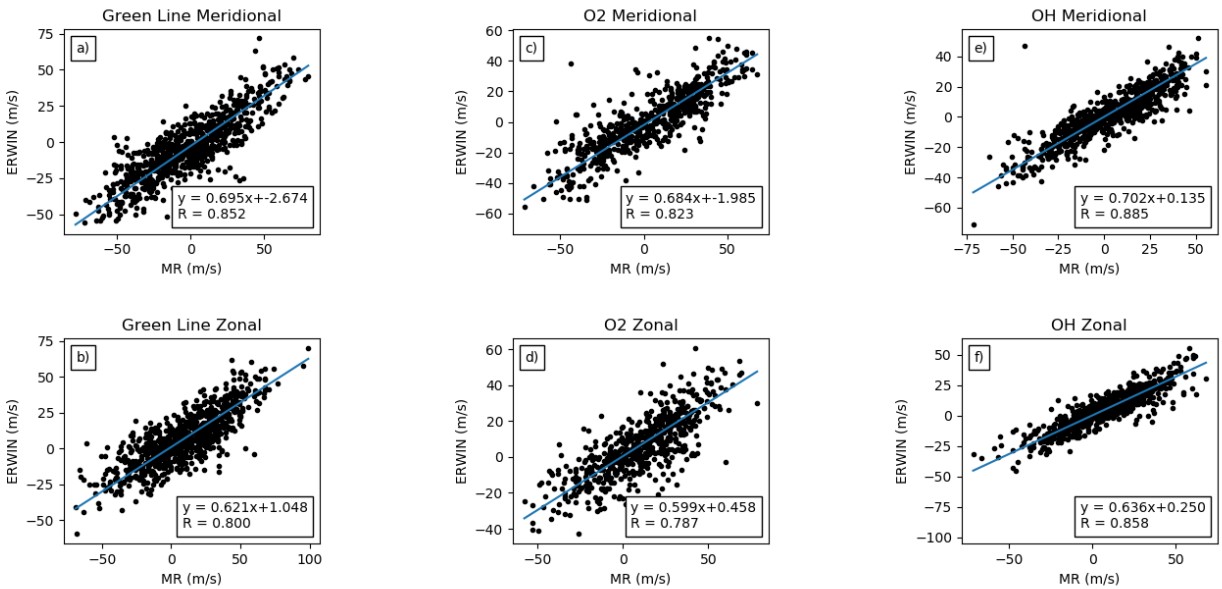

**Figure 10.** Correlation scatter plots of the ERWIN (y-axis) and best height, Gaussian weighted MWR (x-axis) shown in Figure 9. The meridional and zonal winds are shown in the top and bottom panels, respectively. The O($^1$S), O$_2$ and OH winds are shown from left to right.

zero intercept).

The correlations for all approaches are good, confirming that the two techniques view the same wind field. Apart from the meridional O$_2$ wind correlations, the correlation coefficients for the wind comparisons increased as more detailed comparisons were undertaken. The single height correlation coefficients were lowest and correlation coefficients where height variations were included, were highest. At the same time, the parameters of the straight line fit varied between the approaches with no particular approach consistently resulting in "optimal" fits. Consequently, the fit associated with the highest correlation co-

efficient (i.e. the comparisons where the height of best correlation was considered) is taken to be the best indication of the relationship between winds obtained using the two techniques.

    In interpreting these results, several aspects of the comparisons, need to be taken into account. Significant emission for the O$_2$ and O($^1$S) airglow lies above the height range of the MWR. Furthermore, the wind uncertainties of the lowest and highest

MWR height are larger so comparisons involving these heights would be less precise. Both these factors point to the comparisons between the OH winds and the MWR being most representative of the relationship between these two instruments as the OH airglow layer lies within the height region where the meteor radar winds are most precise. This is born out in the observations where the correlations are highest for OH and the offsets close to 0 m/s.



| Emission | Wind Component | Correlation Type | r value | Slope | Intercept (m/s) |
|---|---|---|---|---|---|
| OH | Meridional | a | 0.844 | 0.731 | -0.462 |
| | | b | 0.884 | **0.772** | **0.108** |
| | | c | **0.885** | 0.702 | 0.135 |
| | | | | | |
| | Zonal | a | 0.857 | 0.698 | -1.891 |
| | | b | 0.844 | **0.747** | -1.271 |
| | | c | **0.858** | 0.636 | **0.250** |
| | | | | | |
| $O_2$ | Meridional | a | **0.837** | 0.625 | **-1.222** |
| | | b | 0.823 | 0.663 | -2.877 |
| | | c | 0.823 | **0.684** | -1.985 |
| | | | | | |
| | Zonal | a | 0.774 | 0.702 | -9.759 |
| | | b | 0.765 | **0.712** | -6.458 |
| | | c | **0.787** | 0.599 | **0.458** |
| | | | | | |
| $O(^1S)$ | Meridional | a | 0.848 | **0.767** | -2.884 |
| | | b | 0.849 | 0.725 | **-2.495** |
| | | c | **0.852** | 0.695 | -2.674 |
| | | | | | |
| | Zonal | a | 0.782 | **0.689** | -0.837 |
| | | b | 0.788 | 0.677 | **-0.812** |
| | | c | **0.800** | 0.621 | 1.048 |

**Table 1.** Summary of correlation results for the three different approaches: a - Correlation for best single height for full time series; b - Correlation for best single height for Gaussian fit for the full time series; c - Correlation for best running mean height (2-Day window) and Gaussian fit. Underlined, bold-faced text identify the optimal parameters for each wind direction and emission.





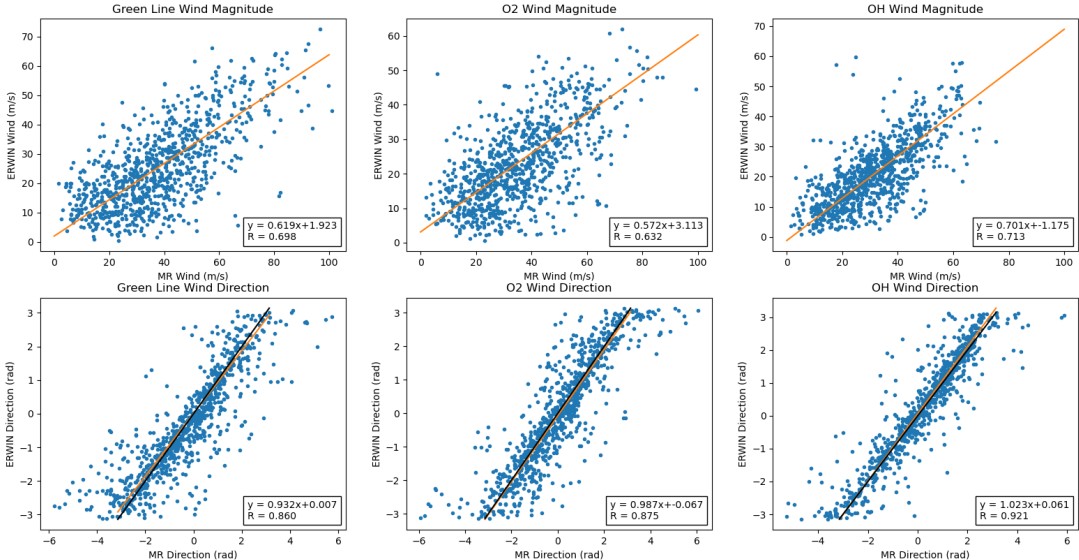

**Figure 11.** Correlation scatter plots of the magnitude and direction of the ERWIN (y-axis) and 2-day Gaussian weighted MWR (x-axis) shown in Figure 9. The magnitude and direction of the winds are shown in the top and bottom panels, respectively. The O($^1$S), O$_2$ and OH winds are shown from left to right.

Two aspects of the linear fits are striking. The first is that the intercepts are close to zero. The second is that ERWIN consistently measures winds that are on average factors of 0.6 to 0.7 of those of the MWR, a result that has also been noted in recent Fabry-Perot, MWR comparisons [Yu et al. (2017); Lee et al. (2021)].

The small values for the intercepts for the OH fits are especially impressive as they are associated with two very different
wind observation techniques. This implies that the zero wind values for both techniques are unbiased and consistent with each other. Zero wind for ERWIN is determined from the daily average vertical wind along with thermal drift monitoring using a spectral calibration lamp (for details, see Kristoffersen et al. (2013)) whereas the MWR uses internal frequencies and 2-D fits to many meteors to set the zero. This implies that both instruments have value for validating satellite winds. The slightly larger zero wind offsets in fits for the O$_2$ and O($^1$S) comparisons is likely associated with the extension of the airglow layer above
the meteor radar observation range. The sign of the wind shear in the upper levels of the MWR mean winds match the sign of the offsets.

Interpreting the difference in wind ratio between the two techniques requires more thought. One would expect the temporal averaging of the ERWIN winds and the spatial averaging of the MWR winds to result in the same data product since the





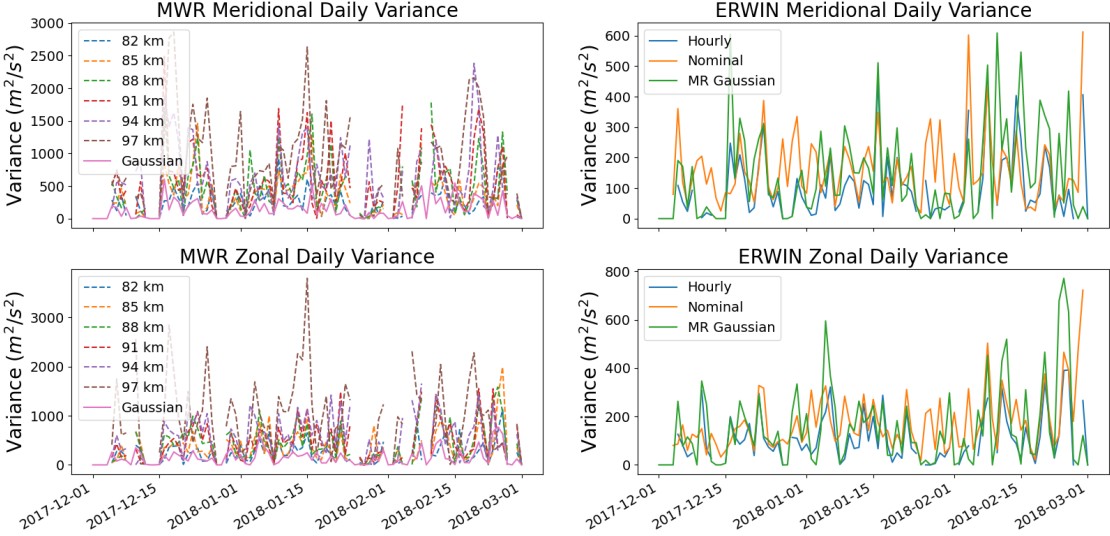

**Figure 12.** Plots of the daily variances for the MWR (left panels) and ERWIN OH winds (right panels) for the meridional (upper panels and zonal (lower panels) components. The MWR panels include time series of the daily average variances at each height as well as that for the Gaussian weighting used in comparisons with the ERWIN hourly averaged winds. The ERWIN panels include time series of daily variances of the nominal winds and hourly averaged winds. For comparison, the Gaussian weighting used in comparisons with the ERWIN hourly averaged winds is also included in these panels

temporal and horizontal wind variations would be averaged out. A slope of ∼1 would be expected. As this isn't the case, the explanation must lie in differences in the instrument filters.

To explore this difference further, the daily variances of the observations from each instrument were calculated (see Figure 12) using the OH emission winds (this emission falls, nominally, inside the MWR range and as noted earlier provides the best

comparison between the two techniques). In this figure, the left panels are for the MWR variances and the right panels the ERWIN variances. The upper panels are for the meridional components and the lower panels for the zonal components. In the MWR panels, time series of daily variances for each height and the optimal Gaussian height weighted averages are plotted. In the ERWIN panels, daily variances of time series calculated using the nominal ∼5 minute cadence ERWIN observations (i.e. prior to the hourly averaging), the 90 minute ERWIN averages and the optimal Gaussian height weighted MWR averages.

The height specific MWR variances at all heights are much larger than the ERWIN nominal variances. This implies that the MWR wind variability at each height is significantly greater than the temporal variability observed in the nominal ERWIN data.




|  | MWR Gaussian Variance ($m^2/s^2$) | MWR Variance Ratio | ERWIN 90-minute Variance ($m^2/s^2$) | ERWIN Variance Ratio |
|---|---|---|---|---|
| Meridional | 209.4 | 2.63 | 111.9 | 1.66 |
| Zonal | 211.8 | 2.41 | 115.3 | 1.52 |

**Table 2.** Variability associated with the optimal height Gaussian vertical averaging of the MWR winds and temporal averaging of the ERWIN OH winds for both wind components. Columns 2 and 4 contain values of the variances for these two wind determination over the full time period. Column 3 contains values of the ratio of the MWR wind variances calculated at the nominal OH layer height (88 km) to the vertically Gaussian weighted wind variance. Column 5 contains the ratio of the variances calculated using the nominal ( 5 minute) ERWIN measurements to the 90-minute averaged ERWIN wind variances

One might expect that this difference would disappear after temporally averaging the ERWIN observations and vertically averaging the MWR observations as this averaging was undertaken to generate equivalently averaged data products from each instrument. However, as can be seen in the right panels of Figure 12, the daily variance associated with the Gaussian averaged time series of winds is generally greater than the daily variance associated with the hourly averaged ERWIN data. This is consistent with the correlation analyses where, as noted earlier, the wind variation in the MWR observations tends to be larger than that from ERWIN.

Table 2 summarizes these results. Variances (column 2) for the full two month time interval for the zonal and meridional winds were calculated for the MWR best Gaussian weighting as determined in Section 4. The ratio of the MWR variance at the nominal OH layer peak location, 88 km, to these MWR best fit variances was also calculated (column 3). Similarly, the variances for the 90 minute averaged ERWIN winds (column 4) over the full time interval were calculated and the ratio of the variances calculated using the nominal ( 5 minute) ERWIN measurements to these 90-minute averaged ERWIN wind variances (column 5). Of note is that although the vertical averaging reduced the variance associated with the MWR winds by a factor of $\sim 2.5$ for both wind components, the resulting variance of $\sim 210 \ m^2/s^2$ is still a factor of 2 larger than the time averaged ERWIN winds ($\sim 113 \ m^2/s^2$). Temporal averaging reduced the variance associated with the nominal ERWIN winds by a factor of $\sim 1.6$. These results further confirm that the height resolved variability in the horizontal wind observed with the MWR is significantly larger than the temporal variability in the horizontal wind observed with ERWIN.

In summary, the observations show that for all three airglow emissions, the wind directions measured by both instruments agree, zero winds agree and the airglow winds are smaller whatever the direction. The proportionality constant is independent of wind magnitude. There is more variability in the radar winds even after vertically averaging to match the airglow layer thickness.





## 6   Discussion

Given that the spatial and temporal averaging does not result in time series of similar variability, there must be another reason for this difference. Both techniques are based on the approximation that the wind field over the observation region is uniform. This assumption is reasonable for longer term averages as long as the gravity wave field is isotropic. For the one hour cadence
time series considered here, this is unlikely to be true for each data point. Non-uniformities in the meteor field in the presence of gravity waves with horizontal scales of the same order or smaller than the radar field of view will result in additional MWR wind variability. However, similar considerations also apply to ERWIN so this is unlikely to be the source of the differences in wind magnitude between the two techniques.

One aspect of the observation process that might affect the instrument filters is the the height distribution of the airglow emission rate. If the airglow distribution isn't symmetric in height but is weighted toward lower heights, the symmetric distributions used in this study might not correctly weight the wind variability (this generally increases with height as a result of conservation of energy associated with waves). The maximum e-folding growth scale of this variability if no dissipation is assumed is twice the scale height or  10-12 km. In this case, weighting issues of  1 km would result in the airglow wind to radar
wind ratio being about 0.9, significantly greater than observed.

Another factor which might cause the observed difference in wind magnitudes is sensitivity to the phase of gravity waves present in the field of view. For example, it is known that airglow brightness is modulated by gravity waves with maximum brightness occurring at the maximum downward displacement of air parcels [Ward (1999)]. For freely propagating gravity
waves the horizontal and vertical winds are directly correlated [Fritts (2000)] for waves with vertical wavelengths $\lambda_z << 4\pi H$ (where H is the local scale height). In this case, the brightness is in quadrature with the horizontal wind. Should the occurrence of meteor trails have any dependence on wave phase similar considerations could also come into play. The essential point here is whether there is a tendency for certain parts of the phase cycle of the gravity waves present in the sampled field to be favoured. This would result in an unequal weighting of the winds during the spatial and temporal averaging associated with
each technique and could explain the observed differences in wind variation between them.

To explore this possibility, a simple model of airglow weighting is developed and the cases with and without weighting compared. The motion field is envisioned to consist of a constant background horizontal wind profile and horizontal winds associated with gravity waves. The Gaussian averaging undertaken earlier with the MWR is essentially an average over the
airglow layer of the height variability of the wind associated with the meteor radar. For uniform weighting, this can be roughly approximated as a box car height averaged wind, $\bar{u}$, with the layer defined as a weighting function with a value of 1 in the layer and zero elsewhere. For airglow weighting, the height variation of the weighting function has a sinusoidal component similar in form to the gravity wave but shifted in phase. This model is also applicable to airglow weighted temperature measurements





with the observable of interest being temperature instead of wind.


Assuming an upward propagating gravity wave, with downward propagating phase fronts, positive horizontal winds to the right, and positive vertical winds upward, the vertical variation in horizontal wind $u(z)$ for a single wave is given as:

$$u(z) = \delta u \cos(\frac{2\pi}{\lambda_z}z + \phi) \tag{1}$$

Here $\phi$ is the phase of the wave at the base of the airglow layer, $\delta u$ amplitude of the horizontal wind variation, $\lambda_z$ is the vertical
wavelength. Assuming a background wind profile, $u_\circ(z)$, after integrating over the layer, the observed wind, $\bar{u}$ is given as

$$\bar{u} = \frac{1}{\Delta z} \int_0^{\Delta z} \left( u_\circ(z) + \delta u \cos\left(\frac{2\pi}{\lambda_z}z + \phi\right) \right) dz = \bar{u_\circ} + \frac{\delta u}{\pi}\frac{\Delta z}{\lambda_z}\sin\left(\frac{\pi \Delta z}{\lambda_z}\right)\cos\left(\frac{\pi \Delta z}{\lambda_z} + \phi\right) \tag{2}$$

The height averaging results in the averaged wave amplitudes tending to zero as $\lambda_z << \Delta z$ through the sinc function behaviour so that the layer averaged background wind, $\bar{u}_\circ$ is observed. This is the effect expected as a result of the height and time averaging of the winds from the two instruments. For $\lambda_z >\sim \Delta z$ the radar height averaging should accommodate the gravity wave variations and for $\lambda_z << \Delta z$ the influences average out.


To explore the effect of airglow brightness weighting for an arbitrary phase relationship between the airglow and horizontal wind, a weighting function is introduced into the averaging. Assuming the same geometry for the gravity wave and form of the horizontal wind perturbation, $u(z)$, as described above, the vertical, relative variation of the associated airglow brightness,
$w_I(z)$ can be expressed as

$$w_I(z) = 1 + \alpha \cos\left(\frac{2\pi}{\lambda_z}z + \phi - \gamma\right) \tag{3}$$

Here $\alpha$ is the ratio of the airglow brightness variation associated with the wave to the average brightness of the layer. $\gamma$ is the phase difference between the phase of the horizontal wind and the airglow weighting of the wind. For the airglow variations relative to the horizontal wind for the non-dissipating gravity wave described above, $\gamma = \frac{\pi}{2}$.


Inclusion of the effect of the modulation of the airglow brightness by the wave results in a height averaged horizontal wind, $\bar{u}^*$, given by:

$$\bar{u}^* = \frac{\int_0^{\Delta z}(u_\circ(z) + \delta u \cos(\frac{2\pi}{\lambda_z}z + \phi))(1 + \alpha \cos(\frac{2\pi}{\lambda_z}z + \phi - \gamma))}{\int_0^{\Delta z}(1 + \alpha \cos(\frac{2\pi}{\lambda_z}z + \phi - \gamma)dz} = \bar{u}_\circ^* + \bar{u}_{gw}^* \tag{4}$$

$\bar{u}_\circ^*$ is the contribution to the averaged horizontal wind associated with the background profile and $\bar{u}_{gw}^*$ is the averaged con-
tribution associated with the wave. The weighted averaged wind associated with the background wind and significant longer period waves (dominantly the semidiurnal tide during polar winter) will be reasonably well approximated by the radar profile. The background profile is unlikely to have significant small scale variations, the vertical gradient is small and the significant



semidiurnal variations have vertical wavelengths significantly larger than the airglow layer [Pancheva et al. (2020)]. In addition, as $\alpha$ is roughly 0.2 for shorter period gravity waves (see for example Figure 2 in Sivjee et al. (1987)), the contribution of the airglow weighting to $\bar{u}_\circ^*$ will be small for waves with vertical wavelengths of the order of the meteor radar resolution and will average to zero as the vertical wavelength decreases.

This is not the case for $\bar{u}_{gw}^*$ since the airglow modulation and horizontal wind are phase related. Setting $\sigma = \frac{\Delta z}{\lambda_z}$, the number of wavelengths in the layer,

$$\bar{u}_{gw}^* = \delta u \left( \frac{\frac{1}{\pi\sigma}\cos(\pi\sigma + \phi) + \frac{\alpha}{4\pi\sigma}\cos(2\pi\sigma + 2\phi - \gamma)\sin(2\pi\sigma) + \frac{\alpha}{2}\cos(\gamma)}{1 + \frac{\alpha}{\pi\sigma}\cos(\pi\sigma + \phi - \gamma)\sin(\pi\sigma)} \right) \tag{5}$$

A difference from the unweighted case is immediately evident from the asymptotic limits of $\bar{u}$ and $\bar{u}^*$ as $\sigma$ approaches zero. In this case, $\bar{u} \to 0$ and $\bar{u}^* \to \bar{u}_{gw}^* \to \delta u \frac{\alpha}{2}\cos(\gamma)$. The bias for vertical wavelengths significantly smaller ($\sigma$ large) than the layer thickness is negative or positive depending on whether the weighting is in phase or out of phase with the wind whereas when the weighting is in quadrature, there is no bias. For waves with large vertical wavelengths compared to the layer thickness ($\sigma$ approaching zero) the phase of both the wind and airglow weighting remains approximately constant throughout the layer so the effect of the weighting becomes negligible.

Figure 13 shows the normalized wind amplitude variation as a function of various model parameters. Here we only include consideration of the $\bar{u}_{gw}^*$ since, as noted above, for small vertical wavelengths, $\bar{u}_\circ^*$ is negligible. Panel a illustrates the normalized layer weighted amplitude variation expected for a uniformly weighted layer (no weighting curves in the figure) as described in Equation 2 as a function of normalized wavenumber, $\sigma$ and phase at the base of the layer, $\phi$. As expected, as $\sigma$ increases, the averaging results in a reduction in the amplitude. Panels b and c illustrate the effect of airglow weighting on the layer average, again as a function of $\sigma$ and $\phi$, for $\gamma = 0$ and $\pi/2$. Note that for $\gamma = 0$ the airglow weighting results in what is effectively a cosine squared term so that the wavenumber dependence is modified and instead of tending to zero as $\sigma$ increases a finite offset remains. For $\gamma = \pi/2$ the higher modulation remains but the offset tends to zero with larger $\sigma$. In both cases there is significant amplitude variation for small $\sigma$ but in the context of the vertical averaging of the radar measurements, this would be accounted for.

Panels d though f present curves showing the variation of the height averaged amplitude as a function of $\sigma$ for different $\phi$: 0, $\pi/2$ and $\pi$ respectively. In each panel, curves for uniform weighting, and airglow weighting for $\gamma = 0$, $\pi/2$ and $\pi$ are plotted. In all three cases, the curve for $\gamma = \pi/2$ almost coincides with the uniformly weighted layer and the curves with $\gamma = 0$ and $\gamma = \pi$ lie above and below these curves respectively. The effect of airglow weighting varies with $\gamma$, with maximum positive shift occurring when the airglow perturbations are in phase with the horizontal wind, maximum negative shift when they are out of phase with the horizontal wind and no shift when they are in quadrature. The shift is positive for $-\pi/2 < \gamma < \pi/2$ and negative for the complementary set of phases.



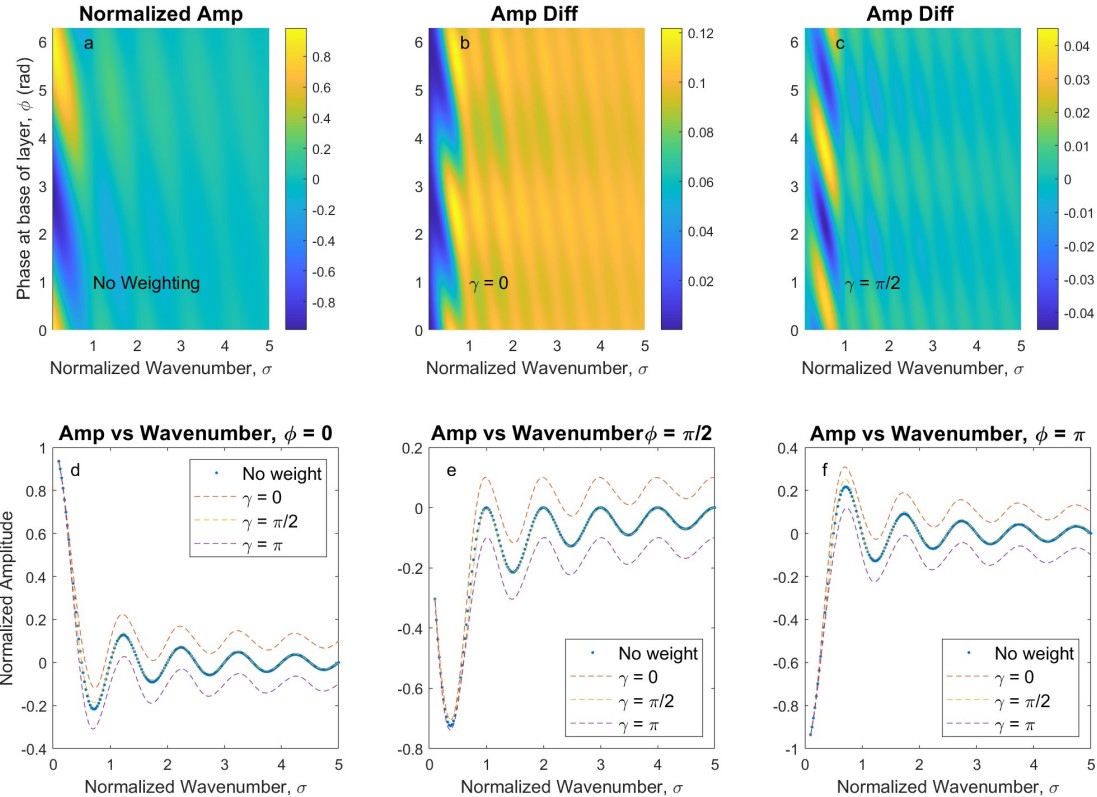

**Figure 13.** Plots illustrating the effect of airglow weighting associated with gravity waves as a function of normalized wavenumber, wave phase at the base of the layer and phase of the airglow weighting relative to the wave phase. Panel a shows the normalized, layer averaged amplitude for the case with no airglow weighting. Panels b and c are plots of the difference between airglow weighted layer average amplitude and the non-weighted case for $\gamma = 0$ and $\pi/2$ radians respectively. Panels d to f are plots of the layer weighted, normalized amplitude for $\phi = 0$, $\pi/2$ and $pi$ respectively. In each of these panels curves for the no weighting case, and $\gamma = 0$, $\pi/2$ and $pi$ are plotted. See text for details.





This analysis shows that small vertical scale gravity wave horizontal variability weighted by phase related airglow can systematically result in deviations from the non-weighted average, proportional to the perturbation horizontal wind. In particular when the phase difference is zero or $\pi$ the respective deviations are positive or negative and when it is $\pi/2$ or $3\pi/2$ they are negligible. Should the radar winds preferentially be weighted so that the detected meteors tended to occur for wind maxima in the waves, the zero phase case could preferentially occur. However, this appears unlikely as recent comparisons between MIGHTI winds and specular meteor radars indicate a slope close to 1 [Harding et al. (2021)] (although the viewing geometry is different from ground based observations).

For airglow weighting to be the cause of the difference in wind magnitude between the radar and the airglow winds, the horizontal wind and airglow brightness must be out of phase. This is possible if dissipating gravity waves with periods significantly less than the radar sampling time are present. This is possible as is indicated in the theoretical work of Vadas and Nicolls (2012) and recent observations of gravity wave characteristics [Nicolls et al. (2012); Lu et al. (2017)]. Lu et al. (2017) in particular (Figure 10), note that shorter period gravity waves in the mesopause region exhibit phase differences indicating significant dissipation and phase differences between temperature and vertical wind approaching $\pi$. For quasi-adiabatic motion, the airglow brightness would be in phase with the temperature and hence out of phase with the horizontal wind).

It is interesting to note that for ground based airglow weighted temperature measurements, discrepancies with satellite temperature determinations have been noted with the ground based temperatures being greater [von Savigny et al. (2004)Oberheide et al. (2006), Parihar et al. (2017), Ammosov et al. (2019)]. Since temperature variations associated with freely propagating gravity waves are in phase with airglow variations, the above analysis (with temperature being considered the observable of interest) implies that ground based temperature observations using airglow would be expected to be higher than satellite based observations as observed.

This model indicates that airglow weighting of horizontal wind could be the cause of differences between the meteor radar and winds derived from airglow. Reductions in the observed wind are feasible for dissipating waves as long as the phase difference between the horizontal wind and airglow is close to $\pi$. This reduction is proportional to the amplitude of the horizontal wind as is observed.

The periods and wavelengths of importance to this analysis remain poorly resolved in current observations [Lu et al. (2017)]. Confirmation of the relevance of this model requires further refinements in observation techniques and model development. Possible aliasing issues and the observation geometry relative to gravity wave phase fronts also need to be considered. However, these would be of particular importance for individual case studies and less important for the statistical analysis undertaken in this paper. The consistency of the form of the correlations and linear fits over all directions and emission used with ERWIN implies that the assumption of directional isotropy of the observed waves is valid. The aliasing and geometrical effects will




contribute to the spread of differences between the two instruments. Exploring the details of this aspect of the comparison is outside the scope of this paper.

## 7 Conclusions

An extensive and robust assessment of meteor radar/ground-based interferometric wind measurements is presented in this pa-
per. This is achieved through comparisons of co-located meteor radar/ERWIN observations from PEARL, Eureka, Nu. Unique to this comparison are ERWIN's world leading measurement accuracy, rapid observation cadence and multiple emission observations. These allow, for the first time, an exploration of the temporal and vertical characteristics of the two techniques.

The best correlations between the wind measurements of the two instruments occurred when winds from each instrument
were averaged to match the spatial and temporal characteristics of the other. This was achieved by vertically averaging the radar winds using a Gaussian envelop to approximate the vertical airglow profile, by temporally averaging the ERWIN winds to match the 90 minute observation interval of the radar, and allowing the peak height of the Gaussian envelop to vary to achieve maximum correlation. These heights matched well with the peaks in co-located SABER airglow observations.

The important conclusions from these comparisons are:

1. The zero winds for both instruments are in excellent agreement despite being based on completely different techniques.

2. Airglow derived wind magnitudes are about 0.7 of the radar winds for all three airglow emission, i.e. independent of emission and wind magnitude. This is in agreement with previous comparisons between meteor radar and Fabry-Perot wind observations using airglow.

3. The wind directions from the two instruments agree very well implying that the wind magnitude ratio mentioned above is independent of direction.

4. The mean height of best Gaussian fit relative to the accepted nominal peak, correlation coefficient, slope of fit and intercept is best for the OH observations. We attribute this to the fact that the OH emission airglow profile is the best match to the height range of radar observations whereas for the other emissions part of the emission profile extends
above the top of the region where precise meteor wind measurements are made.

5. The variability in the radar winds is significantly greater than that in the ERWIN winds. This implies that the height variability in the wind field is significantly greater than the temporal variability for the averaging scales associated with this comparison (90 minutes and $\approx 8$ km).

The excellent zero wind match between the two techniques, implies that it is only motion relative to the ground that is
involved in the biases and hence the instrument filters are the cause. As noted above, it is unlikely that radar winds are biased positively so a mechanism by which ERWIN wins are biased negatively must be found. Possibilities include deviations in



actual airglow profiles from the Gaussian airglow profile used in the averaging and airglow brightness weighting. The former possibility is unlikely to result in the observed ≈0.7 magnitude reduction.

To explore airglow brightness weighting, a simple model of airglow averaging of a wind field composed of a steady background wind profile and gravity waves was developed. The airglow layer was modelled as a boxcar layer, and gravity waves were assumed to involve sinusoidal variations in the vertical of horizontal wind and airglow brightness with a phase difference between them. This demonstrated that unless these field were in quadrature, as the vertical wavelength became small relative to the layer thickness, the airglow weighted mean wind would deviate from the unweighted mean by an amount proportional
to the cosine of the phase difference, the amplitude of the wind perturbation and the magnitude of the airglow modulation. For airglow modulation to cause a a significant reduction in the height averaged horizontal wind, the phase difference would need to be close to $\pi$. This would only be possible for dissipating waves. As noted earlier, there is some observational evidence and theoretical support for wave dissipation for shorter period gravity waves.

Observations of the dynamics and constituent transport in the mesopause region are crucial to understanding coupling between the upper atmosphere/ionosphere and lower atmosphere. Recently ground based radar networks [McCormack et al. (2017), Chau et al. (2021), Stober et al. (2022)] have been used to study three dimensional wind fields at these heights. Other multi-instrument investigations combine radar, airglow images and temperature and satellite observations to better interpret airglow heights [Younger et al. (2015), Reid et al. (2017)] and gravity waves [Vargas et al. (2021)].


The combination of ERWIN type airglow wind and brightness observations with radar winds described in this paper is another example of the scientific richness possible through multi-instrument studies. Insights were gained into the airglow layer height, the nature of the wind variability and instrument functions and the gravity wave spectrum. In the future, a more detailed analysis of the instrument functions will be undertaken to identify the sensitivity of each technique to vertical winds, aliasing
and airglow weighting. The wind imaging capability currently being developed [Langille et al. (2016), Kristoffersen et al. (2022)] will contribute significantly to resolving these questions. Case studies for individual events, for which all sky images and satellite observations can also be included, are planned. This would allow the atmospheric state to be better specified so that details of the observations can be more completely explored. Studies such as these, are important steps in resolving the details of mesopause region dynamics and transport and their role in coupling the upper and lower atmosphere.

*Data availability.*

Data sets for this research are openly available in the University of New Brunswick Libraries DATAVERSE Research Data Repository Ward et al. (2023). The analysis software used in this paper are standard routines for correlation analysis.



*Author contributions.*

All three authors contributed to the collection and analysis of the observations, paper conceptualization and the discussion of the methodology and results. WW was responsible for the development of the airglow model. The manuscript was prepared by SK and WW and all three authors contributed to the revisions.

*Competing interests.*

The authors declare no competing interests .

*Acknowledgements.*  The ERWIN and meteor wind radar measurements were made at PEARL by the Canadian Network for the Detection of Atmospheric Composition Change (CANDAC), which has been supported by the Atlantic Innovation Fund/Nova Scotia Research Innovation Trust, Canada Foundation for Innovation, Canadian Foundation for Climate and Atmospheric Sciences, Canadian Space Agency (CSA), Environment and Climate Change Canada (ECCC), Government of Canada International Polar Year funding, Natural Sciences and Engineering Research Council (NSERC), Northern Scientific Training Program, Ontario Innovation Trust, Polar Continental Shelf Program, and Ontario
Research Fund. Dr. Sam Kristoffersen was partially supported by the NSERC CREATE Training Program in Arctic Atmospheric Science. We thank CANDAC/PEARL/PAHA PI James Drummond, PEARL Site Manager Pierre Fogal, meteor wind radar mentor Alan Manson, CANDAC Data Manager Yan Tsehtik, the CANDAC operators, and the staff at ECCCs Eureka Weather Station for their contributions to data acquisition, and for logistical and on-site support. WW is grateful for a three month visit to the National Institute of Polar Research (NIPR) arranged by Dr. Ejiri and Prof. Nakamura during which a significant portion of this paper was written. Support for this visit was provided by
NIPR and an Eaton Fellowship.





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
