# Peer review of "Wind comparisons between meteor radar and Doppler shifts in airglow emissions using field widened Michelson interferometers"

_EGUsphere, 2023_

## Author Comment (AC1)

We thank this reviewer for their inciteful and comprehensive comments. We have modified the manuscript for the most part in accordance with their suggestions. It has resulted in an improved paper.

Our responses (in red) along with reference to the line numbers of any associated substantial changes to the paper follow.

Reviewer #1 Comments

In this work, wind measurements in the mesosphere-lower thermosphere region between two instruments using very different techniques are compared. One is a meteor radar with large horizontal observational volume; the other is based on interferometry of airglow emission layers that has high horizontal resolution but lower vertical resolution with changing average emission layer height. From three months of data in polar winter, correlation analysis between wind magnitudes and direction have been carried out, subsequently refining the technique of comparing measurements from both instruments. In the most sophisticated, airglow layer height was variable and radar winds were weighted to match a supposed Gaussian emission layer profile. It was found that the wind magnitudes derived from airglow were a factor of 0.7 of the winds derived from the meteor radar. One possible reason for the airglow winds to be underestimated was explored in some detail, i.e. a modulation of the airglow layer brightness in height due to small-scale gravity waves in a way that gravity wave phases associated with lower horizontal wind are favoured. The paper is well written and explained and the work is very valuable as wind is a critical quantity that is difficult to measure in this altitude range. Validating different techniques is very important. I have a number of minor comments that might help to improve the manuscript and try to summarize the main points before giving a list by line numbers.

Correlation coefficients are calculated but a thorough error analysis with significance levels and confidence limits is lacking. The intepretation is often only qualitative, e.g. "significantly larger", "match well", etc. But to interpret the physics of a tiny increase between two correlation coefficients or lack of it might not make sense if both are insignificant. E.g. I would have expected higher R values in Fig. 7 compared to Fig. 6, but there is no significant improvement, rather a decrease. What is the interpretation then? That over two or three months, the airglow layers cannot be approximated by a Gaussian? But shouldn't over a long enough time frame an approximation by a Gaussian become reasonable? Maybe it shouldn't be interpreted at all, because all R values are insignificant, I am not sure. It is also crucial if the observed differences in wind magnitude are significant or not. Uncertainties should be addressed in more detail. It is also relevant to show that the meteor radar sufficiently samples the large observation volume in order to proof that the larger-magnitude winds compared to the ERWIN measurements are not due to inhomogeneous horizontal wind structure and sampling issues with the radar. The number of meteors has to be evaluated for this.

One of the points of this paper is that changes in the manner that the two wind measurements are compared, produce small changes in the correlation coefficients and fits. As the difference between the correlation coefficients is small, we have used the offset and slope of the fit as additional parameters to use to distinguish between comparison approaches.

Additional analysis to include uncertainties in the correlation coefficients, which are typically on the order of 0.01, shows that, with a few exceptions, the three methods provide equal

correlations within the uncertainties. These are now included in Table 1. All correlations are robust and R values significant. The possibility of a null hypothesis is extremely low.

In terms of the Gaussian approximation of the layer, the variability of the layer height over time (shown in Figure 8) will limit the accuracy of the application of a constant Gaussian layer over the entire data set. As shown in the paper, the most accurate measurement of the layer is likely the 2-day running mean of the Gaussian fit shown in Figure 10.

As noted in the description of the meteor radar (Section 2.2), wind measurements with the meteor radar are not provided unless 7 suitable meteor echoes are present. Although the echoes might not be uniformly distributed throughout the volume, any variability will also be present in the ERWIN winds as any non-uniformity is unlikely to match the ERWIN observation points. Our assumption is that for the time-period used for the analysis, that these non-uniformities would affect both measurements similarly.

A more detailed examination of individual echoes and ERWIN winds, in principle is an excellent one, but, as discussed in more detail in our response to Reviewer 2, must be done very carefully and is out of scope for this paper.

There is a mismatch between text and figures regarding the dataset (in l. 63 it is stated that data is from Dec 2017 and Jan 2018. The plots show also data of Feb 2018). Please state clearly how long and large the data set is, i.e. time range, number of hours, number of cloudy days removed. (around l. 170). When the instruments were installed in 2006 and 2008, respectively, and have always been operational, why use only two months of data? Is the used dataset sufficiently large? Why was this time period chosen, and what are the implications? How were conditions, was there a stratospheric warming? A paragraph should be added about the expected background. What are wind magnitudes expected in this region? Do seasonal averages of the wind magnitude and direction match? What are the dominant processes? Is the variability mostly due to tides? I also would encourage to look closer at the data. E.g. in Fig. 12, I wonder why the values of the green "MR Gaussian" line is sometimes larger than the others when the former constitutes a vertical average. Can the variance then truly be larger?

For this paper, the time-period used for the analysis included December 2017 and January and February 2018. The text has been modified to be consistent with this. We chose this period because it was a non-warming year, and both instruments were operating satisfactorily. We consider this to be a representative year with sufficient observations to do robust comparisons between the two instruments.

We have added the following sentence to the paper to describe why this period was chosen (line 66).

"This period was chosen because it was a non-warming year, and both instruments were operating satisfactorily. We consider this to be a representative year with sufficient observations to do robust comparisons between the two instruments."

There have been very few mesopause region wind observations at these latitudes published in the literature so no consensus of what is typical at these latitudes exists. It is a topic that will be examined in more detail in the future. The purpose of this paper is to compare ERWIN and

the meteor radar so that such a comparison involving these two instruments could be done rigorously.

To address this, we included the following sentence and reference to earlier wind observations at Eureka (line 66),

"The observed wind variability and magnitude is in line with earlier wind observations at Eureka [Oznovich et al., 1997; Manson et al., 2011]."

and the following sentence in the conclusions to indicate the future research that this comparison enables (line 567).

"These comparisons provide the foundation for a general study of mesopause winds over a range of scales over Eureka."

The Gaussian average having a larger variability than some of the lower heights can be explained by the increase in wind magnitudes (and therefore variances) with increasing height and the weighting profile that is being used. The variances at lower heights will generally be less than those at higher heights (energy conservation given the density decrease). Given that the Gaussian weighted profile is a weighted average, the variance should be less than some of the layers, but not necessarily all of the layers. In addition, the lowest heights, which are the heights that sometimes have lower variability than the weighted mean, would (typically) have lower weights due to the nominal layer heights (87, 94 and 97 km).

Also, for the meteor radar it might be worth looking into the extreme values e.g. around 15 January - are they truly that large, are they related to some event, or is there some problem with the data?

We do not think that there is an issue with the meteor radar winds around Jan 15. For statistical purposes, we think it is best to simply use the data sets as provided through standard analyses without modifying them. Examining outliers for case studies is something that would be valuable to do in the future.

Regarding the mismatch of wind magnitudes and variances, I have several questions. The authors argue that the averaging should have produced comparable magnitudes and variances, but I am not convinced this is true (l. 348, l. 390). The footprint of the instruments is still different and together with structures of wind below the resolution and unsufficient sampling, this could well result in different variances. Naively I would have expected another outcome, that the ERWIN variances are larger than the meteor radar variances, when the latter are averaged over a larger volume. We know from visual polar mesospheric cloud observations that there is considerable variability on the km scale which likely transfers to the wind field. An uneven spatial sampling due to low meteor numbers might increase the variability of the meteor winds. I also wonder if there is a possible effect due to the horizontal distance due to ERWINs viewing geometry?

We basically agree with the reviewer's point. One of the purposes of the paper is to point out these incongruences and contrast them to what would naively be expected (i.e. similar magnitudes and variances). The wording in our initial draft did not convey this. To ensure that the readers understand that we are dealing with hypotheticals we have changed the wording of lines 348 to 351 to:

*"One might expect the temporal averaging of the ERWIN winds and the spatial vertical averaging of the MWR winds to result in the same data product since the temporal and horizontal wind variations would be averaged out. If this was the case, a slope of ~1 would be expected. As this isn't the case, the explanation must lie in differences in the observational filters for the systematic difference over the full data set."*

*We also agree that the variances between the two techniques as a result of combinations of their observing geometries and particular spatial and temporal wind configurations, may occur. However, we would argue that for this is unlikely to be the case for seasonal averages where particular conditions would be unlikely to persist. Examination of particular events where there is enough known to examine possible systematic effects on the wind observations would certainly be of interest is out of scope for this paper.*

In the discussion of the proposed gravity-wave-phase mechanism I was at first confused about the relation of airglow brightness and horizontal wind. The latter is derived from the Doppler shift and thus independent of the absolute airglow layer brightness. The trick here is, I think, the change of airglow brightness along the imaged column, because only the integral is measured and the height information is lost. If there then is a dependence on gravity wave phase, this could introduce a bias. It is a relatively complicated argument and needs to be explained carefully to the reader. As the argument was extended to the meteor radar, it is not immediately clear to me how the occurrence of meteor trails should depend on the gravity wave phase, please explain. Also, please state explicitly for what type of waves this effect is relevant, i.e. vertical wavelengths below 5 km and horizontal wavelengths between 5 and 60 min (?).

*The reviewer has interpreted our argument correctly. Phase related variations within the integration region of any instrument can affect the associated observable. For ERWIN, this is weighting variations in height within the field of view. For the meteor radar this would be within the horizontal region associated with each height bin.*

*To clarify the airglow weighting argument, we have added the following sentence to the first paragraph dealing with this topic (line 405).*

*"Wave correlated brightness variations associated with waves with vertical wavelengths smaller than the layer width (~8 km) could preferentially weight one wave phase over another and bias the layer integrated wind."*

*To clarify the point regarding the meteor trail potential bias we have added the phrase (line 406):*

*Should the occurrence of meteor trails have any dependence on wave phase "in the horizontal region associated with each height bin (for example if the meteorite ablation occurrence is sensitive to the velocity difference between the meteorite and atmosphere)"*

*To clarify the reason why bias in meteor radar is unlikely we have added the phrase (line 488)*

*"and isotropy of meteor directions and gravity wave directions would average this out when long time periods are considered."*

I suggest to use the term "observational filter" instead of the here used "instrument filter" (l. 94 and later). For the latter, the reader could imagine some filter built physically into an instrument. The meaning intended however is of the capability of a technique to observe only part of the relevant spectrum of the process that is to be studied, e.g. by a limitation of range or resolution. I think this is better expressed as "observational filter".

Thank you for this suggestion. I agree that the term "observational filter" is clearer and more appropriate. All instances of "instrument filter" have been changed to "observational filter".

I suggest a sub-structuring of section 4 regarding the selected methods that are applied to the data before comparison: Around line 193, "The first", "the second" are mentioned, but "the third" is missing. I suggest to make subsection "4.1 Method A" in l. 211, "4.2 Method B" in l. 240, "4.3 Method C" in l. 262 or similar.

Thanks for the suggestion. We have reorganized this section as proposed.

The figures can be improved. Some are hard to read because there is too much data in it such that no good visual comparison is possible (e.g. Fig. 2 and 9), some are redundant and can likely be removed (Fig. 3 and 4), and some can be extended.

In response to this comment, we are removing Figures 3, 4 and 9 from the body of the paper. We think that having an overview figure showing the two data sets is useful so we are keeping Figure 2.

l. 1 "winds in the upper mesosphere" it is clear to an expert that meteor radars measure wind in that altitude range, but it should be mentioned in the beginning for non-expert readers

Added "in the upper atmosphere".

l. 5 "at all three heights" please mention the heights explicitly, maybe in the brackets in line 4

Added the layer heights in the description of the airglow layers (97 km, 94 km and 87 km).

l. 5 Michelson and Fabry-Perot are mentioned. Please make more clear in the abstract how many instruments are used, and what their names and techniques are

We have substantially revised the first 6 sentences of the abstract to address this comment and the next two. We thank the reviewer for querying this as it led to a clearer abstract.

The first part of the abstract is now:

"Upper atmosphere winds from a meteor radar and a field-widened Michelson interferometer, co-located at the Polar Environment Atmospheric Research Laboratory in Eureka, Nu., Canada (80$^\circ$ N, 86$^\circ$ W) are compared.The two instruments implement different wind measuring techniques at similar heights and have very different temporal and spatial observational footprints. The meteor radar provides winds averaged over a $\sim$300 km horizontal area in 3 km vertical bins between 82 and 97 km on a one-hour cadence. The E-Region Wind INterferometer (ERWIN) provides airglow weighted winds (averaged over volumes of $\sim$8 km in height by $\sim$5 km radius) from three nightglow emissions (O($^1$S) (oxygen green line, 557.7 nm, 97 km), an O_2 line (866 nm, 94 km), and an OH

line (843 nm, 87 km)) on a ~5 minute cadence. ERWIN's higher precision  (1-2 m/s for the O(1S) and OH emissions and ~4 m/s for the O_2 emissions) and higher cadence allows more substantive comparisons between winds measured by meteor radar and Doppler shifts in airglow emissions than previously possible for similar meteor radar/airglow Doppler shift comparisons using Fabry-Perot interferometers."

l. 6 "higher accuracy/higher cadence" higher than what? The meteor radar? The past? Please state this clearly

The "higher accuracy/higher cadence" phrase refers to the improved capabilities of the field-widened Michelson interferometer relative to the Fabry-Perot. We have modified the abstract as noted above (response to l. 5 comment) and made this explicit.

l. 8 "airglow and radar winds" I understand the meaning, but more correctly, it is wind measured by means of meteor radar or Doppler shifts of airglow emissions.

As noted in our response to the comments on l. 5, we have modified the abstract substantially and included the term "Doppler shifts of airglow emissions" explicity.

l. 65 please make the difference between Michelson interferometers and Fabry-Perot more clear. Do you refer to previous work/literature that used Fabry-Perot interferometers, and the difference in this work is the use of Michelson interferometers?

The fundamental difference between the two instruments is that the field-widened Michelson interferometers have a significantly larger throughput than the Fabry-Perots and as a result can take more precise wind measurements at a faster cadence. This is now mentioned in the text: "but the former instrument has a significantly larger throughput. Consequently, …" (l 65).

l. 107 how are the accuracies determined?

The accuracies are determined using the statistical standard errors for each viewing direction. This is discussed in more detail in Kristoffersen et al., 2013, a citation to which was added to the text. These statistical uncertainties were found to be in agreement with the theoretical uncertainties based on the line visibility and brightness (also discussed in Kristoffersen et al., 2013).

Fig. 1 could be extended by adding three Gaussians centered at the relevant heights for ERWIN on the left (and the green shading removed) and the height distribution of meteor from the dataset used in 3 km bins on the right. That would make it very clear and helpful.

We have revised the figure as suggested. The caption description is augmented with the following sentence.

"Gaussian profiles to the left and right of the upper figure, indicate the approximate locations of the three airglow layers and the meteor radar echo distribution"

[Figure]

| Instrument | Spatial Bin Size z by radius | Temporal |
|---|---|---|
| Meteor Radar | 3 km x 150 km 1 location | 60 min. Vertical profile |
| **ERWIN** | **8 km x 5 km 5 locations** | **5 min. Three heights** |

l. 123 please make clear what the 5 min cadence refers to. Are measurements in the five directions "simultaneously" (l. 114, that is 5 min of data every 5 min for each direction), or is it "sequenced through" (l. 123)? How much time does the calibration take?

The five directions are measured simultaneously, with each airglow layer being measured sequentially. It takes about 45 s to scan the Michelson mirror through the necessary path differences to provide a measurement of the line-of-sight winds (5 directions from which the meridional and zonal winds are calculated) for each airglow emission. Therefore, with some overhead for calibration etc., the 3 emissions are measured in about 5 minutes. This cycle is repeated, providing three time-series with sampling periods of about 5 minutes. The following was added to the text to help clarify this point: "such that meridional and zonal winds for each airglow emission are measured every five minutes". The calibration takes about the same amount of time as the emission measurements, so about 45 s per wavelength.

l. 136 what are "140 bin quadrants"? Please explain more about how the uncertainties are determined. Do they change with time?

To simultaneously measure the 5 line-of-sight wind directions, a quad mirror with a hole in the centre is used to direct light from each of the LOS wind directions onto a separate section of the CCD. Each of these quadrants mapped onto the CCD contain roughly 140 CCD bins. The uncertainties are determined by taking the standard error of the respective quadrant, with the LOS wind value being the mean. To clarify this, we replaced "140 bin quadrants" with "140 bins per quadrant".

Yes, these uncertainties do change with time; the typical values for clear days, such as those used in this study, are provided in the paper. Error bars were not included in the figures as this would make them too difficult to read.

l. 180 please give the values of the determined variances

The standard deviations are provided in the following figure:

[Figure]

As noted in the paper, these are dominated by the geophysical variances since the wind uncertainty for ERWIN is 1-2 m/s and these are generally an order of magnitude larger.

l. 209 what is meant by "generally complement the MWR winds well"?

Here "generally complement …" means "follow the MWR in direction and relative magnitude. We have changed the text to this phrase.

Fig. 2 does this figure show ERWIN winds filtered with a 90 min running mean? I find the type of visualization with the arrows very difficult. There is so much variation on short time scales, it is impossible to compare the time series. Maybe consider making eight color plots for the radar and the three transmissions and meridional/zonal?

This figure is a plot of the ERWIN winds filtered with a 90-minute running mean. We have included this as part of the description in the caption and text. As we are removing Figure 9,

we think that Figure 2 serves the purpose of providing the reader with a general sense of the agreement between the two instruments.

l. 220 how was the correlation coefficient determined? Please state significance levels and confidence intervals.

The correlation coefficient is the Pearson correlation coefficient, and was determined using the data shown in the figures. Uncertainties are typically 0.01-0.02, and have been added to Table 1.

Fig. 3 and 4. I see little value in these plots, as Fig. 5 summarizes the results. It would be an option to remove them. If kept, significance levels and confidence intervals should be added.

We agree with the reviewer that Figures 3 and 4 can be removed from the paper and we have done so.

l. 229 it is not "roughly Gaussian". Please add the Gaussian fit and parameters in the figure.

"Roughly Gaussian" was a descriptive term used to describe the general shape of the correlation functions. We did not do any fitting here. We think this term is an appropriate description of the shape of the correlation curves and would like to keep it.

l. 237 R=0.77 might not be an excellent correlation.

As noted earlier, the correlations are robust and given the large number of data points going into the correlations the term excellent is appropriate.

l. 266 please motivate the use of a 2-day window. What processes act on these scales?

The 2-day window was chosen as a balance between providing a higher temporal resolution and keeping enough points for the correlation. At this timescale, tides would remain important, but larger scale circulation patterns would act like constant background values. The following text was added to clarify the choice of the 2-day window: "This two-day running correlation was chosen such that a high temporal resolution was obtained, whilst using sufficient data in the correlation."

l. 279 consider adding a line plot of the six time series of emission heights and SABER (in one plot) that might visualize better the agreement ("basic pattern") in addition to the figures of Fig. 8.

The SABER observations are primarily important for the hydroxyl emissions, and we feel that the agreement is clearly illustrated in the lower panels. Although the proposed extra plot could be interesting, we do not feel it provides much new information relative to the current plot.

l. 283 what is meant by "variations of close to +- 5km" and "total variation close to 10 km"?

The variations are the variations in the emission layer height, it has been changed to 'variations in the observed emission layer height' in the text to clarify this.

l. 287 this is a very short paragraph and makes a somehow unfinished expression, as does Fig. 9. It is hard to read anything useful from this plot. I cannot see that the "agreement is good" and "match on average".

To make this paragraph clearer, the text was changed to the following: "The two time-series show similar long-term patterns on scales of a day or larger." We are removing Figure 9 from the paper as it is difficult to interpret as the reviewer has suggested.

l. 320 please re-evaluate whether approaches are good based on significance levels and confidence intervals

Due to the high number of degrees of freedom, the p-values for these figures (even Figure 3 with some small correlation coefficients, e.g., -0.094) are below 0.01, which indicates that there is a low statistical probability of a result at least as extreme based on the null hypothesis. Given this, it seems that the correlation coefficients may provide a more precise measure of the statistical correlation of the data.

l. 329 "wind uncertainties" please add the magnitude of the uncertainties in the respective earlier section

We have added the uncertainties to the section on the meteor radar (line 155).

"The uncertainty is greater at 82 (~7 m/s) and 97 km (~12 m/s), where fewer meteors are detected."

l. 361 "is significantly" -> "is about an order of magnitude"

"significantly" changed to "about an order of magnitude".

l. 361 remove "temporal" in "the temporal variability"

"temporal" was removed.

l. 361 please add an interpretation to this finding. I guess it makes sense as ERWIN implies a vertical average. The conclusion then would be that there is significant variability on vertical scales between 3 km (radar vertical resolution) and about 5 km (ERWIN vertical resolution)?

We agree that this implies significant variability in vertical scales between those resolvable by the meteor radar and those resolvable by ERWIN. We have included the phrase "for vertical scales between those resolvable by the meteor radar (~ 2 km) and those resolvable by ERWIN (~7 km)" at the end of this paragraph.

l. 363 Again, I am not convinced that comparable variability can be expected due to the averaging.

As noted above, we agree that there might be systematic differences in the variability for the two techniques. However, the expectation that the averaging should eliminate the differences is a reasonable initial assumption against which the actual results are contrasted.

l. 366 the magenta line of Fig. 12 left panels could be added to the right panels for direct comparison as the y axis have different scales.

Yes, the green line labelled 'MR Gaussian' in the right panels is the same as the magenta 'Gaussian' line in the left panels.

l. 378 is this sentence true? It is not the height-resolved MR winds against the temporal variability of the ERWIN winds. It is the vertically averaged MR wind against the (inherently) vertically averaged ERWIN winds.

Yes, there is a larger decrease in the variability when comparing the height averaged MWR results compared to the decrease in variability when considering the ERWIN temporal averages. This suggests that there is more variability on the 3 to 10 km heights scales than in the 5 to 90 minute time scales.

l. 429 what is meant by "For lambdad_z >~ Delta z the radar height averaging should accomodate the gravity wave variations"? In that case, there will be a bias?

The point here is that as the vertical wavelength of the wave becomes larger than the airglow layer thickness, the vertical resolution of the radar should resolve the wave and no bias will result. To indicate this clearly, we have replaced "radar height averaging" with "vertical resolution of the radar observations" and included the sentence "In both cases no bias is present." at the end of the paragraph.

l. 447 "The background profile is unlikely to have significant small scale variations" doesn't that depend on the definition of the background profile? It was defined to be constant in l. 413, so by design it does not have small scale variations. Also with real data, it depends on the separation into perturbations and background how smooth the background is.

The background wind profile is assumed to be temporally constant over the observation period of ERWIN and can vary with height. The separation into a background and gravity waves means that the background consists of a superposition of tidal, planetary and any other long period waves, and the zonal mean. Given the typical characteristics of these waves, we would not expect significant small scale vertical variability. To clarify this we have changed the sentence (line 413-414) to:

"The motion field is envisioned to consist of a temporally constant background horizontal wind profile (including the zonal mean and longer period waves such as tides and planetary waves) and horizontal winds associated with gravity waves."

Eqn. 4 Can you give expressions for the two summands on the right. It is then easier to understand how the first summand depends on alpha as argued in line 450.

The expression for the second summand is already given in equation 5. The first summand is $\frac{1}{D} \int_0^{\Delta z} u_o(z) \left( 1 + \alpha \cos\left( \frac{2\pi}{\lambda_z} z + (\phi - \gamma) \right) \right) dz$ where D is the expression in the denominator of equation 4. The exact dependence depends on the form of $u_o(z)$ and the vertical wavenumber. Assuming a wind gradient of the form βz, the numerator has the form $\beta (\Delta z)^2 \left\{ \frac{1}{2} + \frac{\alpha}{(2\pi\sigma)^2} \left( 2\pi\sigma \sin(2\pi\sigma + (\phi - \gamma)) - \cos(2\pi\sigma + (\phi - \gamma)) - \cos(\phi - \gamma) \right) \right\}$ where

$\sigma = \frac{\Delta z}{\lambda_z}$. As noted, for large σ, any airglow variability will average out and for small σ, the airglow variation will be close to uniform over the airglow layer so the uniform weighting condition is recovered. For $\sigma = 1$, $\bar{u}_o^* = \beta(\Delta z)\left(\frac{1}{2} + \frac{\alpha}{2\pi}\sin(\phi - \gamma)\right)$. $\bar{u}_o^* = \frac{\beta(\Delta z)}{2}$ when there is no airglow weighting. Hence, the maximum relative deviation due to the airglow weighting is $\frac{\alpha}{\pi} \sim .06$ for $\alpha = 0.2$.

The text in the vicinity of Line 450 has been modified by inserting the equation describing $\bar{u}_o^*$ and the condition for $\sigma = 1$ in line with the above discussion.

l. 510 "The periods and wavelengths of importance" please give the values, are they periods above 5 min (the measurement cadence) and below 1 h (the radar temporal resolution) and vertical wavelengths below 5 km (the width of the airglow layer), is this correct?

Yes, this is basically correct (though the meteor radar winds are 90-minute average). We have modified the text by inserting the following text "(i.e. periods between ~5 and 90 minutes and vertical wavelengths less than ~7 km)" in line 510.

l. 512 what is meant by aliasing issues, please explain.

There may be aliasing of waves with smaller wavelengths than the distance between the ERWIN observation locations. The following has been inserted into the sentence:

"(associated with waves with horizontal wavelengths smaller than the horizontal distance between the ERWIN observation locations)".

l. 512 "with observation geometry relative to gravity wave phase fronts" do you mean the tilted beams?

Yes, the issue here is that the averaging along the ERWIN line-of-sight will be sensitive to the tilt of the gravity wave wavefronts. We have modified the text as follows:

"… ERWIN line-of-sight observation geometry relative to gravity wave phase front tilts"

l. 521 "world leading measurement accuracy" this should be shown with the relevant citations of comparable instruments in the introduction or second section.

The Kristoffersen et al paper of 2013, provides such a comparison and is referenced in the second section. Another comparison is included in Kristoffersen et al., 2022 (See Figure 17), We have now included this reference after the quoted text. **

l. 545 "As noted above, it is unlikely that radar winds are biased positively" Where was this noted? Please repeat the argument. Why is it unlikely?

See the comment above on page 4 of our response where we augment the initial discussion on this topic. and isotropy of meteor directions and gravity wave directions would average this out when long time periods are considered. We have added the following at line 545 to remind the reader of this earlier discussion, "(i.e. longer-term isotropy and ICON comparisons"

Minor changes:

l. 355 "In this figure, …" two sentences can be removed

Yes, given that this is described in the figure caption it is superfluous here. These two sentences have been removed.

l. 349 "spatial averaging" -> "vertical averaging"?

"spatial" changed to "vertical"

l. 414/415 this is a complicated sentence, please rephrase.

This sentence has been rephrased as follows: "The Gaussian averaging undertaken earlier is an average of the height variability of the wind associated with the meteor radar over the airglow layer."

l. 423 Delta z is not defined

Thanks for catching this. This is the airglow layer thickness. Text has been added to define this term.

l. 464 there are no panels a, b and c labeled in Fig. 13

These panels are labelled in the upper left corner of each panel. The font size of the labelling was quite small and has now been increased.

l. 532 add "(Fig. 10)" as reference for the value 0.7

Added a reference to Figure 10.

Fig. 6 please add the selected altitudes to the figure labels or caption

The heights of best correlation in Figure six were added to the caption: "The heights of best correlation are a) 94 km, b) 94 km, c) 94 km, d) 91 km, e) 88 km, and f) 88 km."

Fig. 8 it is hard to see the difference between the magenta and red crosses

Thank you for this comment, the figure has been changed to be easier to read. The colour map has been changed to a grey scale, the blue x's are now blue circles, the red +'s are now red triangles and the magenta +'s are now green triangles. The corresponding caption and text in the body of the paper has been changed to reflect the updated figure.

Fig. 8 why are the top of the profiles 0.0 (green)? Is there not data? Then please consider using white color.

There were a few times (particularly at high altitudes) when there were no data. Given that the grey scale colour map has white as 1, these values were left grey denoting a correlation of 0.

Fig. 12 "nominal winds": please add "(at 5 min resolution)"

Added "(at 5 min resolution)" to the caption.

Definition of abbreviations: l. 3 "ERWIN" is not defined at this stage; l. 62 "MWR" not defined. "Meteor wind radar winds" is double, maybe just "MR winds"?; l. 87 "MLT" not defined.

We added a definition of ERWIN in the abstract, and "hereinafter referred to as MWR" for the meteor radar. Added "Mesosphere and Lower-Thermosphere (MLT)" so that MLT is defined.

Typos: l. 1 "two" is double; l. 10 0.3 m/s; l. 147 double "in"; l. 216 missing word: ", and all those"?; l. 239 please add units to the values for slopes and intercept; l. 316 missing word "cardinal wind direction"?; l. 395 double "the"; l. 413 verb is missing; l. 553 "fields"; l. 556 double "a"

l. 1 second "two" was removed; l.10 corrected units; l.147 second "in" was removed; l.216 added "and"; l.239 added units for the intercepts (the slopes are unitless [(m/s)/(m/s)]); l.316 added "cardinal"; l.395 second "the" was removed; l.413 this line seems correct to me; l.553 changed "field" to "fields"; l.556 removed second "a".

A copy editor may help with hyphens and comma, I just give some examples:

Use of hyphens: wind-measuring instruments, field-widened, airglow-weighted, gravity-wave-airglow-brightness-weighting (?, maybe better change that sentence..), gravity-wave-associated-wind, layer-weighted, space-based, ground-based, multiple technique -> multi-technique, airglow-radiance-weighted, three-body, …

We will look into this with the copy editing.

Comma: l. 11: On average, …; l. 39 Often, …; l. 114 remove comma; l. 167 remove comma; l. 235 "Figure 6, contains" remove comma, l. 250 "the height, and thickness" remove comma; l. 328 "comparisons, need" remove comma; l. 544 "two techniques, implies" remove comma; l. 573 "these, are" remove comma

Commas removed or inserted as suggested.

**Citation**: https://doi.org/10.5194/egusphere-2023-2369-RC1

---

## Author Comment (AC2)

We thank this reviewer for their inciteful and valuable comments. We have modified the manuscript for the most part in accordance with their suggestions. Their comments have resulted in improvements to our paper.

Our responses (in red) along with reference to the line numbers of any associated substantial changes to the paper follow.

Reviewer #2 Comments

"Wind comparisons between meteor radar and Doppler shifts in airglow emissions using field widened Michelson interferometers" by Kristoffersen et al. deals with a comparison between two completely different wind measurement techniques in the mesosphere and lower thermosphere region. Their approach is more advanced than other existing comparison studies between meteor and optical methods by introducing the effect of airglow brightness weighting, which I have long been interested in but have had no opportunity to test myself. I enjoyed very much reading the manuscript and do not have a major concern about what have been done in the present study.

However, the obtained results are still not conclusive and there could be more to be done. I would like the authors to even step forward and to think about the following approach additionally, which will simplify the comparison setting and provide further insight into the difference of the two techniques with less restriction. A major obstacle that makes the comparison difficult is the difference in temporal and spatial resolution/averaging between the two techniques. Although the authors use 90 min averaged and some hundred-km horizontally averaged winds in this study, I would directly use the 5 min ERWIN winds and instantaneous meteor echo radial velocity without conducting such temporal and spatial averaging in order to avoid/minimize unwanted effects caused by small scale gravity waves described in the discussion and conclusion.

The number of meteors detected within the FOV of the ERWIN may be small, but by using meteors within an area such as 10 degree diameter centered around the ERWIN FOV the number will be enough to make a direct comparison with only a minimal spatial averaging because you have two month long data set. Some minor correction of radial wind values for meteor echoes will still be necessary considering the elevation and azimuthal angles of meteor echoes. Further, a frequency spectral analysis of ERWIN radial velocities for individual four FOVs will also be what should be tried to see how the spectral values behave, especially in the high frequency range. Because of their large off-vertical angles most contribution to the radial winds are from horizontal winds and the effect of vertical winds to the spectra will be negligible.

I strongly recommend that the authors try such approach and make the present study even more attractive to potential readers.

 We appreciate the suggestion. This is a good idea.

The suggestion of comparing individual meteor Doppler data with single (one FOV) ERWIN measurements has been tried but was not included in the paper.  It shows statistically similar results (e.g. speed bias) to the current fit comparisons. But there are caveats to be considered:

Since the emission layer heights are nominal rather than measured, the effective sample range (and height) for individual N, E, S, and W ERWIN values is unclear, so even if there were a surplus of concurrent meteor echoes in or near an ERWIN field of view, it would not be possible to choose the "appropriate one".

In addition, the meteor echo is a point measurement, whereas the optical Doppler is a sort of average over a long slant path through an unknown emission layer, which may also include significant height variations in tidal winds.

We felt that these results did not augment the results reported on in this paper. We may try to identify a time period with enhanced meteor echoes and attempt, a study as suggested, but this would be the topic of a new paper.

We have noted the possibility of undertaking a study such as that suggested in a new paragraph in the conclusions.

"Unfortunately, the results of this study do not fully resolve the reason for the difference between winds measured by the two instruments although it does specify the nature of the difference more precisely than previously possible. Future studies can be directed toward attempting direct correlation of individual meteors collocated with the ERWIN observations for periods with enhanced meteors. The airglow model can be refined further by including more realistic airglow profiles and exploring the scale dependence of wind deviations (including the form of the background profile and gravity wave characteristics) due to gravity wave modulation of the airglow brightness. An effect not considered in this paper, is the possibility that scattering might affect optical wind measurements as suggested by Harding 2017. Since indications of the presence of cloud are available at PEARL, a correlative study examining the wind deviations as a function of cloud can be undertaken in the future."

MINOR COMMENTS

Abstract

 line 3

   ERWIN is not defined.

 A definition of the ERWIN acronym was added to the abstract.

 line 10

   0.3/s > 0.3 m/s

 Corrected

lines 85-86

"They consist of ..... cross section"

The meaning is not clear. Grammatical correct?

We have replaced this somewhat condensed sentence with the following to clarify the observation process.

"They are determined from Doppler shifts in airglow emissions integrated along each line-of-sight. The contribution from each point in the field of view is weighted by the airglow volume emission rate at that point."

line 138

"Kristoffersen et a. (2021)" should be

[Kristoffersen et al., 2013]

Citation changed to correct citation.

line 143

one "]" is missing.

Added "]"

line 147

one "in" is redundant.

Removed second "in".

lines 166-167

"[ ]" is missing.

Added parentheses around citations.

Figure 2

Information on direction is missing.

Added the following to the caption to clarify the wind directions shown in the plot: "such that northward is the positive y-direction, and eastward is the positive x-direction".

Figure 8

Red and magenta are confusing.

White can be a better choice for one of them.

Thank you for the suggestion, the figure was updated to help make the symbols clearer. The colour map was changed to a grey scale. The blue x's were changed to blue circles, the red +'s were changed the red triangles, and the magenta +'s were changed to green triangles.

lines 342-343

 "2-D fits to many meteors to set the zero"

 The meaning is not clear.

The line-of-sight Doppler shift for the meteor radar is determined directly from the time shift between the returning signals and the transmitted signal. Zonal and meridional winds are then determined in a least mean squares sense from the echoes in each time/height bin (for details see Hocking et al., 2001). We have changed the text to:

"MRW winds are determined from Doppler shifts and echo directions by least squares fits to horizontal components of radial velocities (see Hocking et al., 2001 for details)"

Figure 12

 It is hard to tell apart the lines in the left panels.

 We agree but are not sure what to do about this. We will think about how to resolve this question before the next submission of the paper.

Equation 2

 delta_z is not defined.

We have added the definition in the text, "(vertical thickness, $\Delta z$) " (line 425).

 delta_z/lambda_z in front of sin is reversed if my calculation is correct.

Yes, thanks for catching this. We have corrected this expression in the text.

 Equation 4

 dz is missing in the numerator.

 One ")" is missing in the denominator.

 Thank you for catching these typos, the dz was added to the numerator and the ")" was added to the denominator.

lines 447-448

 A conjunction is probably missing.

To clarify this sentence, we have added "since" to link the clauses.

"The background profile is unlikely to have significant small-scale variations since the vertical gradient is small …"

line 456

I believe that "sigma approaches zero" should be "sigma approaches infinity" or "lambda_z approaches zero". If not, the description in this line is not immediately evident to me.

Thanks for catching this. You are correct. We have changed this to "sigma approaches infinity".

Figure 13

What alpha and delta_u are used for the evaluation?

In the figure, we are showing normalized amplitudes, i.e. $\delta u = 1$. This is stated in the figure and text. We have noted that the value of $\alpha$ used in this figure is 0.2 in the text and in the figure. Thanks for pointing out that this was missing.

The letters "a,b,c,d,e and f" are too small and hard to recognize.

The letters have been changed to a larger font to make them visible.

Figure caption: "pi" should be in Greek.

Two instances of "pi" changed to "$\pi$"

line 482

Lots of expressions such as "small vertical scale gravity wave horizontal variability" are seen in the manuscript. They are not so easy for a non-native English speaker like me to follow, being vague in what is the subject/object or adjective. Expressions with some moderate use of prepositions would be preferable.

We have started going through the paper to identify and modify these. We will continue to do this and hopefully the copy editor will also help with this.

**Citation**: https://doi.org/10.5194/egusphere-2023-2369-RC2

---

## Author Response (AR2)

Dear Professor Chau,

We have corrected the items noted by Referee 1 in our paper, namely missing whitespaces at the end of sentences (".T" in l.3, l. 78, l. 268, l. 319, l. 600 in the manuscript), and two surplus whitespaces "( 5" in Tab. 2 caption and l. 383). We also revised the sentence (l. 66) to "are from a three-month period between December 2017 and February 2018" to clarify the meaning.

We appreciate the rigorous review of this paper. It made for an improved paper.

Best regards,

William Ward